# KRADA: Known-region-aware Domain Alignment for Open-set Domain Adaptation in Semantic Segmentation

**Chenhong Zhou**[*]                                         *20482795@life.hkbu.edu.hk*
*Department of Computer Science, Hong Kong Baptist University*

**Feng Liu**[*]                                              *feng.liu1@unimelb.edu.au*
*The University of Melbourne*

**Chen Gong**                                                *chen.gong@njust.edu.cn*
*The Key Laboratory of Intelligent Perception and Systems for*
*High-Dimensional Information of Ministry of Education,*
*School of Computer Science and Engineering, Nanjing University of Science and Technology*

**Rongfei Zeng**                                             *zengrf@swc.neu.edu.cn*
*Northeastern University*

**Tongliang Liu**                                            *tongliang.liu@sydney.edu.au*
*Sydney AI Centre, The University of Sydney*

**William K. Cheung**                                        *william@comp.hkbu.edu.hk*
*Department of Computer Science, Hong Kong Baptist University*

**Bo Han**[✉]                                               *bhanml@comp.hkbu.edu.hk*
*Department of Computer Science, Hong Kong Baptist University*

**Reviewed on OpenReview:** *https://openreview.net/forum?id=5II12ypVQo*

## Abstract

In *semantic segmentation*, we aim to train a pixel-level classifier to assign category labels to *all* pixels in an image, where labeled training images and unlabeled test images are from the *same distribution* and share the *same label set*. However, in an open world, the unlabeled test images probably contain *unknown categories* and have *different distributions* from the labeled images. Hence, in this paper, we consider a new, more realistic, and more challenging problem setting where the pixel-level classifier has to be trained with labeled images and unlabeled open-world images—we name it *open-set domain adaptation segmentation* (OSDAS). In OSDAS, the trained classifier is expected to identify unknown-class pixels and classify known-class pixels well. To solve OSDAS, we first investigate which distribution that unknown-class pixels obey. Then, motivated by the goodness-of-fit test, we use *statistical measurements* to show how a pixel *fits* the distribution of an unknown class and select highly-fitted pixels to form the *unknown region* in each test image. Eventually, we propose an end-to-end learning framework, *known-region-aware domain alignment* (KRADA), to distinguish unknown classes while aligning the distributions of known classes in labeled and unlabeled open-world images. The effectiveness of KRADA has been verified on two synthetic tasks and one COVID-19 segmentation task.

---

[*]CHZ and FL contributed equally to this paper.
[✉]BH is the corresponding author.
Our source code is available at https://github.com/chenhong-zhou/KRADA

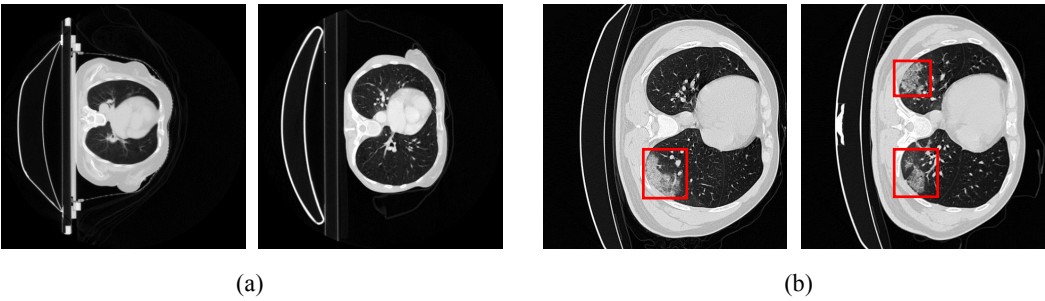

Figure 1: Illustration of the differences in two CT datasets. (a) Examples of normal CT scans. (b) Examples of COVID-19 CT scans, where the infected area is circled by red boxes. These two datasets vary at the visual level (domain shift) and are not consistent at the semantic level (category shift).

# 1    Introduction

Semantic segmentation aims to assign one category label to each pixel in an image, which has a large variety of applications from autonomous driving (Cordts et al., 2016; Siam et al., 2018), indoor navigation (Silberman et al., 2012) to medical image analysis (Tajbakhsh et al., 2020). In recent years, deep learning-based methods have been developing rapidly and achieved remarkable successes in semantic segmentation (Garcia-Garcia et al., 2017; Roth et al., 2022). These methods usually train a *deep convolutional neural network* (DCNN) using a training set that contains pairs of images and pixel-level labels (Long et al., 2015) to segment unlabeled test images. These works commonly assume that the training images and test images are taken from *the same scenario* and share *the same category label set* (Panareda Busto & Gall, 2017; Saito et al., 2018b).

However, in an open world, test images might be taken from a different scenario and have additional category labels compared to training images. For example, in autonomous driving, to reduce the great demand for accurately annotated images, a synthetic dataset, such as SYNTHIA (Ros et al., 2016), is commonly used to train a network for the segmentation of urban scenes. Unfortunately, a realistic urban scenario is quite complex and different from the simulated one. Thus, real-world urban images probably contain some additional category labels (i.e., *unknown classes*) that are not present in the synthetic images.

Another representative example is *Coronavirus Disease 2019* (COVID-19) infection segmentation task. Due to scarce annotated images in COVID-19 datasets, existing chest *Computed Tomography* (CT) scans are expected to be utilized as training images to assist COVID-19 segmentation. However, the segmentation models trained on a normal CT dataset usually show poor performance to segment the COVID-19 infected area, as a result of *domain shift* and *category shift* issues. *Domain shift* refers to a distributional discrepancy caused by the variations in light, conditions, and device types for the acquisition of training and test images, while *category shift* means the inconsistent label sets between training and test images, e.g., COVID-19, a new disease absent in training images. Figure 1 illustrates both issues in the COVID-19 segmentation task.

Regarding such a realistic and challenging segmentation scenario, we name it *open-set domain adaptation segmentation (OSDAS)*. Although *closed-set domain adaptation segmentation* (CSDAS) methods (Hoffman et al., 2016; Vu et al., 2019; Tsai et al., 2018; Luo et al., 2019; Wang et al., 2020; Saito et al., 2018a; Mei et al., 2020; Zou et al., 2019; Zhang et al., 2019; Kang et al., 2020) have been extensively studied to overcome the domain-shift issue, they are not applicable to OSDAS because they probably mistakenly align unknown target data (i.e., open-world images) with source data (i.e., training images), leading to negative transfer (Bucci et al., 2020).

In this paper, to solve OSDAS, we first explore the inherent property of unknown classes and propose that the probability distribution of a given input belonging to the unknown class outputted by a known-class classifier would conform to a prior probability distribution of the known classes. Motivated by the goodness-of-fit test, we propose to use *statistical measurements* to describe how likely a target pixel is an unknown pixel, i.e., how well the output probability distribution of a target pixel *fits* the distribution of an unknown class. The distribution disparity between these two distributions can be measured by statistical measurements. Here we

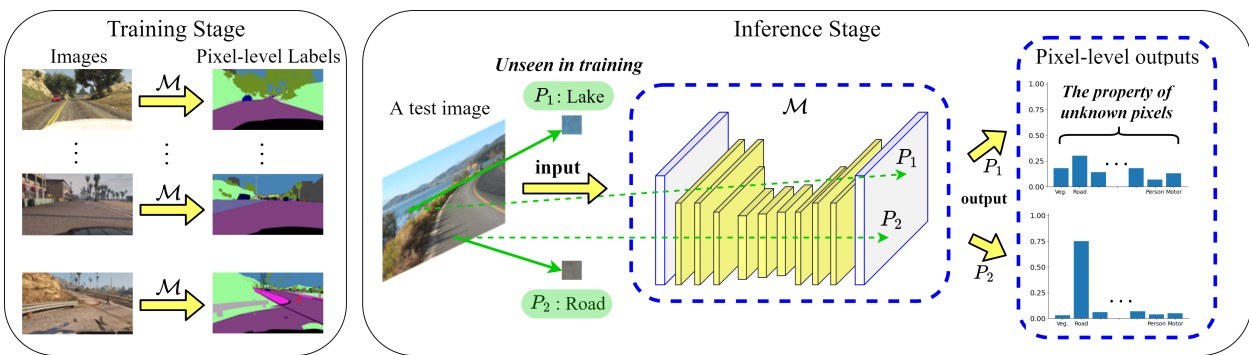

Figure 2: An illustration of the output distribution property of an unknown pixel. A test image contains a "lake" region whose semantic label (lake) is not present during training. The well-trained segmentation model $\mathcal{M}$ *cannot* recognize such a "lake" pixel ($P_1$) and outputs a probability vector over known classes for it. Its probability distribution conforms to a known-class prior probability distribution, different from that of a known pixel ($P_2$).

adopt two statistical metrics: 1) $L_2$-norm (Ahmad & Cerrito, 1993); 2) Kullback–Leibler (KL) divergence (Song, 2002), which are general criteria for testing goodness of fit.

Based on these statistical measurements, we can identify the highly-fitted pixels as "unknown", while the *unknown region* in a target image can be determined. Hence, a segmentation model can be trained using source data and pseudo-labeled target data to achieve a better domain alignment by rejecting unknown target regions and aligning the distributions only for known-class data. We call this framework *known-region-aware domain alignment* (KRADA), which is independent of the network architecture and can be easily realized on existing CSDAS methods to adapt them for OSDAS tasks.

We have realized KRADA on three CSDAS methods in our experiments and evaluated them on two synthetic-to-real street scene segmentation tasks and one COVID-19 segmentation task. Experimental results show that KRADA enables CSDAS methods to identify unknown-class regions and achieve a better overall adaptation, verifying its effectiveness and good generalization ability.

## 2   Problem Setup: Open-set domain adaptation segmentation (OSDAS)

We address the problem of open-set domain adaptation segmentation (OSDAS) that is defined as follows.

**Problem 1** (Open-set domain adaptation segmentation). *Suppose that a set of source images with annotations are denoted as $\{X^S, Y^S\}$ where the source label space is $\mathbb{L}^{H \times W}$ and $\mathbb{L} = \{1, \ldots, K\}$ is the category label set with $K$ known classes. The target images $X^T$ are drawn from a different distribution and the target label set has an additional label $K + 1$ to denote the unseen classes that do not appear in $\mathbb{L}$. We aim to train a segmentation model $\mathcal{M}$ to accurately classify each pixel in target images $X^T$ into one class of the label set $\{1, \ldots, K, K + 1\}$.*

Different from the open world setting in (Bendale & Boult, 2015; Joseph et al., 2021) which entirely concentrates on unknown classes, our proposed setting further considers a common phenomenon in known classes that a distributional discrepancy exists, apart from detecting unknown classes. Moreover, some existing studies about open set segmentation (Hwang et al., 2021; Cui et al., 2020; Cen et al., 2021) also do not consider the distribution shift problem. Instead, we expect the segmentation model in OSDAS to solve both domain shift and category shift issues simultaneously.

## 3   How Do We Determine Unknown Pixels?

To solve OSDAS, the critical issue is how to separate unknown pixels from known pixels in target images. Due to lack of unknown-class supervision, a classifier trained with source data will forcibly label an unknown

pixel as one of the known (source) classes (Boult et al., 2019). To avoid this issue, we consider introducing unknown-class pseudo-labels into a segmentation model so that the model can be trained to learn unknown-class information from pseudo-labels and gain the ability to identify unknown pixels. Before the generation of unknown-class pseudo-labels, we need to know what property an unknown class has. In other words, what can we use to represent an unknown class? If given this property, we can exploit it to determine whether a pixel is unknown.

As unknown classes have never appeared in the training data and there is only prior knowledge of source classes, a well-trained model hardly recognizes an unknown pixel and tends to output a prediction probability distribution conforming to the class prior probability of source classes (Figure 2). Hence, the distribution for an unknown pixel also conforms to such a known-class prior probability distribution. Those target pixels with this property would be considered unknown. In Figure 2, the output of a "lake" pixel (P1) in a test image conforms to a known-class prior probability distribution, as this pixel is an unknown pixel for $\mathcal{M}$.

## 4 A General Framework to Solve OSDAS

Generally, a segmentation network $\mathcal{M}$ consists of a feature extractor $F$ and a pixel-level classifier $C$ to encode input images into the feature space and map the features to the label space, respectively. The network $\mathcal{M}$ can be formalized as $\mathcal{M} = C \circ F$ and it is usually trained with labeled source data:

$$L_{seg}^S = \hat{\mathbb{E}}[\ell(C(F(X^S)), Y^S)], \tag{1}$$

where $\mathbb{E}[\cdot]$ denotes the expectation over m.r.v.s, $\hat{\mathbb{E}}[\cdot]$ is the empirical estimation of $\mathbb{E}[\cdot]$, and $\ell(\cdot, \cdot)$ is a *cross-entropy* (CE) loss function. In OSDAS, an open-set pixel-level classifier $C$ is expected to assign one category label from the label set $\{1, \ldots, K, K+1\}$ to each pixel in target images. However, the known-class supervision $L_{seg}^S$ is not sufficient to make $\mathcal{M}$ both identify unknown target pixels and classify known-class pixels well. Thus, we consider adding auxiliary supervision $L_{seg}^T$ by introducing *unknown-class pseudo-labels* into the training stage:

$$L_{seg} = L_{seg}^S + \alpha L_{seg}^T, \tag{2}$$

where

$$L_{seg}^T = \hat{\mathbb{E}}[\ell(C(F(X^T)), \hat{Y}^T)], \tag{3}$$

and $\hat{Y}^T$ denotes the unknown-class pseudo-labels of target data, and $\alpha$ is a hyperparameter to control the loss weight of unknown classes. Consequently, the open-set classifier $C$ can be trained using the labeled source and pseudo-labeled target data. We introduce the generation of unknown-class pseudo-labels as follows.

### 4.1 Unknown-class Pseudo-label Generation

Based on the distribution property of unknown classes, we consider determining whether a target pixel is unknown from a statistical point of view. Specifically, goodness-of-fit tests indicate the goodness of fit of a model by comparing the observed data with the data expected under the model (Kéry & Royle, 2015; Vasicek, 1976). Motivated by this, we utilize statistical measurements to measure how well the output probability distribution for a target pixel fits a known-class prior distribution. Then, those highly-fitted target pixels would be labeled as unknown pixels.

**Statistical measurements.** A straightforward way to determine unknown-class pseudo-labels is to compare the maximum softmax probability (MSP) with a threshold (Luo et al., 2020). If a pixel whose maximum softmax probability is smaller than a predefined threshold, this pixel will be considered unknown and assigned to $K+1$.

In this paper, we adopt $L_2$ norm and Kullback–Leibler (KL) divergence to measure the distribution disparity between the output probability distribution of a target pixel and a known-class prior distribution. $L_2$ norm, or Euclidean norm, is commonly used to evaluate the discrepancy between two distributions:

$$L_2(p(x), q(x)) = ||p(x) - q(x)||_2 = \left( \sum_{x \in X} |p(x) - q(x)|^2 \right)^{1/2}. \tag{4}$$

In addition, KL divergence is a non-symmetric measure that quantifies how much one distribution differs from another one, a criterion used in goodness-of-fit tests (Song, 2002) to indicate the information lost when $q(x)$ is used to approximate $p(x)$:

$$D_{KL}(p(x)||q(x)) = \sum_{x \in X} p(x) \log \left( \frac{p(x)}{q(x)} \right). \tag{5}$$

**Remark 1.** *Typically, $p(x)$ represents a "true" distribution or a theoretical distribution while $q(x)$ represents the observed distribution. Here $p(x)$ is a known-class prior distribution, and $q(x)$ is the prediction probability outputted by a model for a target pixel. Other statistical metrics can be chosen, and more available metrics are offered in (Gibbs & Su, 2002).*

**Known-class classifier and mark unknown-class pseudo-labels.** To obtain the prediction probability over known classes (i.e., $q(x)$ in Eq. (4) and Eq. (5)), we additionally introduce a pixel-level known-class classifier $C^*$ to output an image defined on the label space. $C^*$ can be supervised by minimizing the segmentation loss on the source data:

$$L^*_{seg} = L^{S,*}_{seg} = \hat{\mathbb{E}}[\ell(C^*(F(X^S)), Y^S)]. \tag{6}$$

For a target image $x^T \in X^T$, $C^*$ aims to produce a probability map $p^{T,*}$ of shape $K \times H \times W$:

$$p^{T,*} = \text{softmax}(C^* \left( F \left( x^T \right) \right)). \tag{7}$$

More specifically, $p^{T,*}_{ij} \in \mathbb{R}^K$ implies the probability vector for a pixel $\{i, j\}$ in $x^T$, and thus $\sum_{c \in \mathbb{L}} p^{T,c}_{ij} = 1$. Here $p^{prior} \in \mathbb{R}^K$ is used to denote a prior probability of known classes, which is a distribution of classes among all of the source data. Hence, we can calculate the distributional divergence between $p^{T,*}_{ij}$ and $p^{prior}$ for each target pixel and compare it with a threshold $\delta$. Those target pixels whose distribution is close to a known-class prior distribution are marked as unknown, which can be formalized as

$$\hat{y}^{T,K+1}_{ij} = \begin{cases} 1, & \text{if } L_2(p^{prior}, p^{T,*}_{ij}) < \delta, \\ 0, & \text{otherwise}, \end{cases} \tag{8}$$

where $L_2$ norm can be replaced with KL divergence or other probability metrics in (Gibbs & Su, 2002).

**Remark 2.** *Note that $\hat{y}^T$ is initially a $(K + 1) \times H \times W$ shaped tensor with all elements equaling zero. $\hat{y}^T_{ij} \in \mathbb{R}^{K+1}$ is a one-hot vector for a target pixel. If $\hat{y}^{T,K+1}_{ij}$ equals one, it means that this pixel will be pseudo-labeled as unknown; otherwise, this pixel remains unlabeled.*

**Adaptive threshold.** We use an exponential moving average method to define the threshold $\delta$ to make it adaptive and smooth by considering the past historical information, denoted as Eq. (9). $\beta$ is a momentum factor to retain the past threshold information. $\Gamma(x_t^S, \gamma)$ indicates the update information from the current source image $x_t^S \in X^S$ at time $t$, as shown in Eq. (10). $|x_t^S|$ indicates the number of pixels in the source image $x_t^S$, and $\gamma$ is a predefined proportion to represent the tolerable ratio of pixels in $x_t^S$ wrongly recognized into the unknown class. As shown in Eq. (11), we sort the $L_2$ norm distance between $p^{prior}$ and $p_t^{S,*}$ in a descending order, where $p_t^{S,*} = \text{softmax}(C^* \left( F \left( x_t^S \right) \right))$. We acquired the threshold information from source images under the given tolerable proportion to adaptively adjust the threshold.

$$\delta_t = \beta \delta_{t-1} + (1 - \beta)\Gamma(x_t^S, \gamma). \tag{9}$$

$$\Gamma(x_t^S, \gamma) = \mathbb{P}_{x_t^S} \left[ \gamma |x_t^S| \right]. \tag{10}$$

$$\mathbb{P}_{x_t^S} = \text{sort}(L_2(p^{prior}, p_t^{S,*}), \text{descending}). \tag{11}$$

Eventually, we can produce pseudo-labels $\hat{Y}^T$ for target images $X^T$. Particularly, pseudo-labels are constantly updated when target images are fed into the network once again. As a consequence, the network can be optimized using source data and previously pseudo-labeled target data while generating new pseudo-labels, which is different from the multi-round training mechanism in (Mei et al., 2020; Zou et al., 2019; 2018) that alternatively optimizes pseudo-label generation and network training.

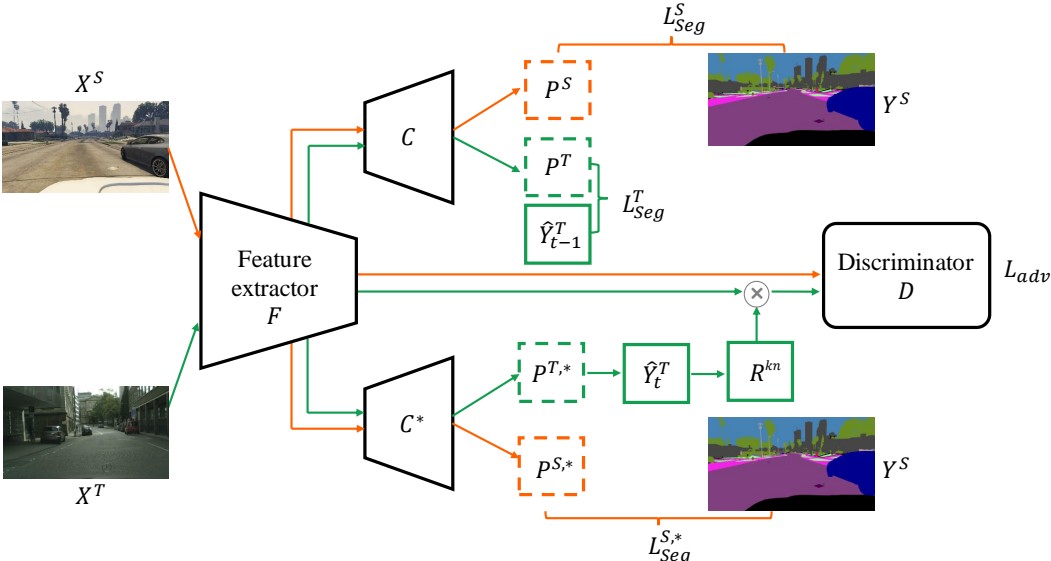

Figure 3: Overview of the proposed KRADA realized on the existing CSDAS methods. It consists of a feature extractor ($F$), a open-set pixel-level classifier ($C$), a known-class classifier ($C^*$), and a discriminator ($D$). The orange and green parts denote the source domain flow and target domain flow, respectively. Specifically, $C$ is optimized under the supervision from both the source domain and pseudo-labeled target domain, where $\hat{Y}_{t-1}^T$ denotes the pseudo-labels of target data previously produced at last time ($t-1$). The aim of $C^*$ is to generate the new pseudo-labels $\hat{Y}_t^T$ at time ($t$) and known-class region $R^{kn}$. Then we forward the features of source images and known regions in target images into $D$ to perform known-class-aware domain alignment.

**Remark 3.** *It should be emphasized that here $L_2$ norm or KL divergence plays a different role from entropy since entropy measures the prediction uncertainty and only depicts the internal relationships of known classes for a target pixel. It does not exploit the distribution property of the unknown class, so it does not measure the distance of a pixel to an unknown class in a strict sense. This is confirmed by the fact that those pixels with high entropy are probably boundary pixels, not unknown pixels.*

### 4.2 Known-region-aware Domain Alignment (KRADA)

Since there is a domain shift between source and target domains, we also consider reducing this domain gap when minimizing Eq. (2). Adversarial training (AT) based methods are a predominant stream of minimizing the domain gap, e.g., existing CSDAS works. These methods train $\mathcal{M}$ to learn domain-invariant features by confusing a domain discriminator $D$. An adversarial loss is minimized to align the distributions between source and target domains at input level (Hoffman et al., 2018; Gong et al., 2019), feature level (Hoffman et al., 2016; Vu et al., 2019; Chen et al., 2019; Wang et al., 2020; Hoffman et al., 2018), or output level (Luo et al., 2019; Saito et al., 2018a; Tsai et al., 2018). The main difference between these three kinds of domain alignment is the input of the discriminator $D$. For a clear illustration, we take the feature-level alignment as an example in the following. In this case, the inputs of the discriminator $D$ are the source features and target features produced by the feature extractor $F$, and the adversarial loss is formulated as:

$$L_{adv}^{AT} = -\hat{\mathbb{E}}\left[\log(D(F(X^S)))\right] - \hat{\mathbb{E}}\left[\log(1 - D(F(X^T)))\right].$$ (12)

Unfortunately, these CSDAS works cannot be directly applied to OSDAS since they would forcefully match the feature distributions of two domains, which makes unknown target data mistakenly aligned with source data, leading to negative transfer.

To avoid this issue, we consider that cross-domain adaptation should be performed only on the known-class data. Therefore, a novel *known-region-aware domain alignment* (KRADA) is proposed to align the target image regions predicted as known with the source images, which is shown in Figure 3. The adversarial loss

---

**Algorithm 1** An implementation of KRADA on existing CSDAS methods.

---

**Input:** source data $(X^S, Y^S)$, target data $X^T$, initial pseudo-labels $\hat{Y}_0^T$.
**Parameter:** network parameters: $\theta_F, \theta_C, \theta_D, \theta_{C^*}$, the number of iteration $N$, learning rate: $\mu$, pseudo-label loss
    weight: $\alpha$, initial threshold: $\delta$, tolerable proportion: $\gamma$, momentum factor: $\beta$.
**Output:** predicted target labels: $\widetilde{Y}^T$.

1: **for** $t = 1$ **to** $N$ **do**
2:     calculate $L_{seg}$ using $(X^S, Y^S, X^T, \hat{Y}_{t-1}^T)$ according to Eq. (1), (2), and (3).
3:     calculate $L_{seg}^*$ using $(X^S, Y^S)$ according to Eq. (6).
4:     generate $\hat{Y}_t^T, R^{kn}$ using $X^T$ according to Eq. (7), (8), and (14).
5:     calculate $L_{adv}$ using $(X^S, X^T, R^{kn})$ according to Eq. (13).
6:     $\theta_F = \theta_F - \mu \bigtriangledown_{\theta_F} (L_{seg} + L_{seg}^* - L_{adv})$
7:     $\theta_C = \theta_C - \mu \bigtriangledown_{\theta_C} L_{seg}$
8:     $\theta_D = \theta_D - \mu \bigtriangledown_{\theta_D} L_{adv}$
9:     $\theta_{C^*} = \theta_{C^*} - \mu \bigtriangledown_{\theta_{C^*}} L_{seg}^*$
10:    update $\delta$ according to Eq. (9), (10), and (11).
11: **end for**
12: Prediction: $\widetilde{Y}^T \leftarrow C(F(X^T))$.

---

can be developed from Eq. (12) to:

$$L_{adv} = -\hat{\mathbb{E}} \left[ \log(D(F(X^S))) \right] - \hat{\mathbb{E}} \left[ \log(1 - D(F(X^T) \cdot R^{kn})) \right],  \qquad (13)$$

where $R^{kn}$ is a binary mask to denote the known-class region predicted for target images. Once pseudo-labels $\hat{Y}^T$ are determined, $R^{kn}$ can also be obtained by:

$$R_{ij}^{kn} = \begin{cases} 0, & \text{if } \hat{y}_{ij}^{T,K+1} = 1. \\ 1, & \text{otherwise.} \end{cases}  \qquad (14)$$

We multiply $R^{kn}$ and each channel of target feature maps and then forward the features of known-class regions in target images into $D$. Eventually, unknown-class regions are rejected, and only the known regions of target images are aligned with source images. It is worth mentioning that KRADA does not have any requirement for the discriminator, so the alignment process of KRADA has no difference from that of original AT methods, except for the target inputs of the discriminator. This indicates that KRADA is *independent of* the model or structure and has good extensibility.

## 5   Realizations of KRADA on Existing CSDAS Methods

In this section, we realize the proposed KRADA on existing CSDAS methods, as illustrated in Figure 3. The original CSDAS network generally consists of a feature extractor ($F$), a pixel-level classifier ($C$), and a discriminator ($D$). To solve OSDAS, we additionally introduce a known-class classifier ($C^*$) which is arranged in parallel with $C$. The architecture of $C^*$ is similar to that of $C$ except for the last convolutional layer with $K$ output channels. The training details are summarized in Algorithm 1. We use source data $(X^S, Y^S)$ and target data $X^T$ with previously generated pseudo-labels $\hat{Y}_{t-1}^T$ to calculate $L_{seg}$ according to Eqs. (1), (2), and (3) (line 2), where the initial pseudo-labels $\hat{Y}_0^T$ is a tensor of $(K+1) \times H \times W$ with all elements being zero.

Then we use source data to calculate $L_{seg}^*$ according to Eq. (6) (line 3). After generating the new pseudo-labels $\hat{Y}_t^T$ and the known-region map $R^{kn}$ according to Eqs. (7), (8), and (14) (line 4), we calculate $L_{adv}$ according to Eq. (13) (line 5). These network parameters and the threshold are updated until the network converges (lines 6-10). Finally, we obtain the segmentation results $\widetilde{Y}^T$ of target data (line 12). Overall, KRADA has no specific requirements for CSDAS architectures ($F$, $C$, and $D$) and can be easily integrated into CSDAS methods to form unified segmentation models. Therefore, a series of CSDAS methods can be adapted to solve OSDAS by a minor modification. In our experiments, we realize KRADA using *three* representative CSDAS methods, and the results show that KRADA can help address the OSDAS problem well.

Table 1: Results on SYNTHIA → Cityscapes. "B" denotes the best score during training, while "L" denotes the last score at the end of training. Source-only refers to a base model only trained on source images without adaptation.

| Method | | road | side. | build. | fence | pole | sign | veg. | sky | person | rider | car | motor | bike | unk. | mIoU | mIoU* |
|---|---|---|---|---|---|---|---|---|---|---|---|---|---|---|---|---|---|
| Soure-only | B | 39.9 | 19.2 | 65.4 | 0.0 | 22.4 | 2.0 | 63.8 | 70.2 | 47.8 | 14.0 | 47.9 | 7.5 | 26.2 | 0.0 | 30.4 | 32.8 |
| | L | 34.5 | 18.0 | 61.7 | 0.0 | 20.9 | 2.0 | 60.1 | 67.8 | 48.0 | 15.7 | 46.5 | 8.4 | 26.2 | 0.0 | 29.3 | 31.5 |
| OSBP (Saito et al., 2018b) | B | 31.7 | 17.4 | 67.5 | 0.0 | 20.3 | 0.4 | 63.0 | 71.0 | 33.3 | 11.7 | 60.8 | 7.7 | 26.2 | 1.7 | 29.5 | 31.6 |
| | L | 28.7 | 16.3 | 66.6 | 0.1 | 19.5 | 0.6 | 59.0 | 72.7 | 28.9 | 7.0 | 50.2 | 3.2 | 23.9 | 2.0 | 27.0 | 29.0 |
| AdaptSegNet (Tsai et al., 2018) | B | 72.4 | 35.7 | 75.6 | 0.0 | 14.4 | 1.1 | 69.9 | 75.5 | 37.7 | 14.4 | 70.6 | 12.2 | 29.9 | 0.0 | 36.4 | 39.2 |
| | L | 66.6 | 35.9 | 74.0 | 0.0 | 14.2 | 1.1 | 68.4 | 72.8 | 35.2 | 13.6 | 62.4 | 10.4 | 27.8 | 0.0 | 34.5 | 37.1 |
| CLAN (Luo et al., 2019) | B | 83.4 | 37.6 | 76.8 | 0.0 | 21.9 | 2.7 | 77.8 | 78.7 | 49.9 | 17.7 | 80.1 | 12.5 | 28.6 | 0.0 | 40.6 | 43.7 |
| | L | 82.9 | 38.1 | 76.1 | 0.0 | 21.7 | 2.3 | 77.1 | 77.0 | 47.5 | 17.1 | 78.5 | 10.7 | 26.8 | 0.0 | 39.7 | 42.8 |
| FADA (Wang et al., 2020) | B | 84.5 | 39.0 | 79.0 | 0.0 | 27.2 | 1.4 | 83.0 | 73.6 | 38.3 | 13.3 | 75.6 | 5.1 | 38.6 | 0.0 | 39.9 | 43.0 |
| | L | 82.6 | 37.5 | 79.0 | 0.0 | 25.8 | 1.8 | 82.9 | 74.8 | 37.7 | 12.9 | 76.3 | 6.6 | 35.0 | 0.0 | 39.5 | 42.5 |
| AdaptSegNet + KRADA | B | 74.1 | 31.5 | 76.5 | 0.0 | 16.3 | 0.7 | 75.1 | 75.9 | 47.9 | 16.0 | 74.4 | 9.8 | 34.6 | 6.1 | 38.5 | 41.0 |
| | L | 66.6 | 30.8 | 75.9 | 0.0 | 17.2 | 0.7 | 74.9 | 75.3 | 46.7 | 16.7 | 73.0 | 9.6 | 36.9 | 5.1 | 37.8 | 40.3 |
| CLAN + KRADA | B | 82.4 | 37.3 | 76.4 | 0.0 | 22.1 | 2.5 | 76.6 | 77.8 | 49.9 | 18.5 | 72.4 | 15.3 | 28.9 | 5.3 | 40.4 | 43.1 |
| | L | 80.2 | 38.3 | 76.9 | 0.0 | 20.3 | 2.5 | 76.8 | 78.8 | 51.0 | 17.7 | 71.6 | 14.2 | 27.9 | 4.8 | 40.1 | 42.8 |
| FADA + KRADA | B | 85.3 | 41.5 | 80.3 | 0.0 | 28.9 | 0.9 | 82.7 | 78.5 | 42.6 | 12.8 | 80.1 | 8.5 | 42.7 | **6.5** | **42.2** | **45.0** |
| | L | 84.6 | 40.0 | 79.9 | 0.0 | 27.6 | 1.5 | 82.4 | 77.4 | 42.6 | 12.7 | 78.5 | 8.5 | 40.5 | **6.4** | **41.6** | **44.3** |

## 6 Experiments

**Synthetic OSDAS tasks.** Since the research about OSDAS has not been explored, there is no public dataset for such a new setting. Based on two synthetic-to-real benchmark tasks in CSDAS: SYNTHIA (Ros et al., 2016) → Cityscapes (Cordts et al., 2016) and GTA5 (Richter et al., 2016) → Cityscapes, we adjust these two tasks to simulate the OSDAS scenario. For the task SYNTHIA → Cityscapes, we select three classes (wall, light, and bus) to form the unknown class and discard those images containing either of the three classes. The remaining images in SYNTHIA are regarded as the source domain. For the task GTA5 → Cityscapes, we choose two classes (fence and sign) to form the unknown class. Those images which do not contain the unknown class in GTA5 are retained as the source domain. Following the common practice in CSDAS, we use the Cityscapes training set as the target domain and evaluate our models on the Cityscapes validation set with a widely adopted evaluation metric: mean Intersection over Union (mIoU) for all classes. Considering the evaluation protocol in unsupervised open set domain adaptation (UOSDA) (Panareda Busto & Gall, 2017; Saito et al., 2018b), we also report the mIoU averaged over known classes only, denoted as mIoU* in this paper. To justify the generalization ability of KRADA, we implement KRADA on three CSDAS methods: 1) AdaptSegNet (Tsai et al., 2018), 2) CLAN (Luo et al., 2019), and 3) a state-of-the-art adversarial training-based method—FADA (Wang et al., 2020), denoted as AdaptSegNet + KRADA, CLAN + KRADA, and FADA + KRADA, respectively. Additionally, OSBP (Saito et al., 2018b) is used as the baseline, which is one of the few UOSDA methods that can be modified to segmentation tasks by directly convolutionalizing its network architecture. More data descriptions and implementation details are described in Appendix B.

**Results on synthetic OSDAS tasks.** We present the results of SYNTHIA → Cityscapes in Table 1. Compared with Source-only, OSBP increases unknown-class IoU from 0% to around 2%. But this achievement is at the cost of the segmentation performance on known classes as both mIoU and mIoU* of OSBP significantly decrease. Compared to Source-only, AdaptSegNet, CLAN, and FADA promote the adaptation for known classes by a large margin, but their unknown-class IoU values are stable at 0%, which means that they cannot recognize the unknown class. After the realizations of KRADA, these three modified models consistently outperform OSBP with huge margins and achieve significant improvements in the segmentation of unknown regions. Compared to their corresponding original versions, these three models equipped with KRADA significantly improve the unknown-class IoU from 0% to around 5% meanwhile obtaining higher mIoU and mIoU* at the last epoch. This is because KRADA mitigates negative transfer caused by unknown classes in the target domain wrongly matched to known classes in the source domain. Thus, the segmentation of known classes also improves. Particularly, FADA + KRADA achieves the best performance at both best and last epochs.

Table 2: Results on GTA5 → Cityscapes. "B" denotes the best score during training, while "L" denotes the last score at the end of training.

| Method | | road | side. | build. | wall | pole | light | terrain | veg. | sky | person | rider | car | truck | bus | train | motor | bike | unk. | mIoU | mIoU* |
|---|---|---|---|---|---|---|---|---|---|---|---|---|---|---|---|---|---|---|---|---|---|
| Soure-only | B | 80.0 | 6.8 | 74.4 | 16.2 | 25.6 | 27.9 | 77.2 | 15.7 | 72.5 | 52.4 | 19.5 | 70.5 | 16.2 | 17.1 | 0.9 | 11.0 | 0.3 | 0.0 | 32.5 | 34.4 |
| | L | 77.6 | 4.3 | 70.9 | 14.9 | 22.0 | 26.5 | 75.6 | 10.2 | 71.3 | 51.8 | 17.0 | 69.6 | 14.9 | 16.3 | 0.1 | 12.4 | 0.3 | 0.0 | 30.9 | 32.7 |
| OSBP (Saito et al., 2018b) | B | 84.9 | 36.8 | 77.9 | 18.4 | 24.1 | 23.0 | 80.6 | 26.3 | 74.5 | 51.3 | 13.2 | 74.8 | 20.1 | 23.0 | 0.0 | 16.4 | 0.0 | 0.5 | 35.9 | 38.0 |
| | L | 85.7 | 35.7 | 78.0 | 15.6 | 23.8 | 20.6 | 79.6 | 29.8 | 70.5 | 51.0 | 11.9 | 72.6 | 18.4 | 19.1 | 0.0 | 16.6 | 0.0 | 0.2 | 35.0 | 37.0 |
| AdaptSegNet (Tsai et al., 2018) | B | 78.3 | 14.4 | 77.9 | 14.4 | 25.6 | 34.3 | 80.2 | 18.3 | 82.9 | 56.0 | 25.0 | 76.5 | 14.7 | 3.8 | 0.5 | 27.9 | 0.9 | 0.0 | 35.1 | 37.2 |
| | L | 71.2 | 14.4 | 73.9 | 10.2 | 24.9 | 33.2 | 81.3 | 19.1 | 83.8 | 53.8 | 24.0 | 72.7 | 14.4 | 1.9 | 0.6 | 30.0 | 1.4 | 0.0 | 33.9 | 35.9 |
| CLAN (Luo et al., 2019) | B | 87.8 | 17.5 | 76.8 | 22.4 | 23.0 | 26.6 | 82.6 | 30.0 | 80.0 | 54.8 | 16.6 | 83.6 | 35.7 | 43.2 | 0.0 | 26.7 | 0.2 | 0.0 | 39.3 | 41.6 |
| | L | 87.2 | 16.7 | 76.0 | 19.9 | 21.7 | 27.3 | 82.5 | 28.4 | 79.2 | 55.0 | 9.3 | 83.1 | 32.2 | 38.0 | 0.0 | 26.8 | 0.2 | 0.0 | 38.0 | 40.2 |
| FADA (Wang et al., 2020) | B | 91.6 | 45.3 | 83.8 | 37.5 | 31.0 | 29.8 | 85.8 | 37.8 | 87.2 | 61.7 | 30.4 | 86.2 | 34.4 | 47.7 | 0.0 | 21.2 | 2.0 | 0.0 | 45.2 | 47.8 |
| | L | 91.5 | 44.5 | 83.9 | 36.5 | 31.2 | 28.1 | 86.1 | 41.0 | 87.1 | 62.0 | 33.1 | 86.3 | 29.2 | 40.3 | 0.0 | 19.8 | 2.0 | 0.0 | 44.6 | 47.2 |
| AdaptSegNet + KRADA | B | 70.5 | 25.8 | 80.1 | 21.4 | 23.8 | 28.1 | 82.9 | 33.0 | 78.1 | 55.1 | 20.2 | 81.7 | 27.2 | 35.9 | 0.5 | 18.1 | 0.1 | 0.7 | 38.0 | 40.1 |
| | L | 71.9 | 25.5 | 79.4 | 21.1 | 23.7 | 26.7 | 82.4 | 31.6 | 78.8 | 53.9 | 18.8 | 79.9 | 27.7 | 35.7 | 1.2 | 16.6 | 0.0 | 0.4 | 37.5 | 39.7 |
| CLAN + KRADA | B | 85.8 | 19.5 | 79.3 | 20.1 | 25.0 | 28.0 | 83.5 | 35.2 | 79.8 | 54.6 | 21.3 | 82.4 | 32.1 | 37.8 | 0.0 | 23.7 | 0.2 | 0.9 | 39.4 | 41.7 |
| | L | 87.2 | 17.5 | 78.4 | 19.7 | 25.0 | 27.6 | 82.9 | 35.2 | 79.4 | 55.7 | 20.5 | 82.3 | 30.2 | 38.0 | 0.0 | 24.2 | 0.2 | 0.6 | 39.1 | 41.4 |
| FADA + KRADA | B | 91.9 | 42.9 | 84.2 | 35.6 | 31.8 | 32.0 | 86.3 | 36.8 | 87.3 | 61.9 | 33.6 | 86.1 | 34.5 | 43.9 | 0.0 | 34.6 | 2.7 | **0.9** | **45.9** | **48.6** |
| | L | 91.7 | 43.6 | 84.4 | 36.0 | 32.0 | 33.8 | 85.9 | 36.2 | 86.8 | 61.6 | 32.6 | 86.0 | 31.0 | 42.0 | 0.0 | 34.3 | 3.1 | **0.8** | **45.7** | **48.3** |

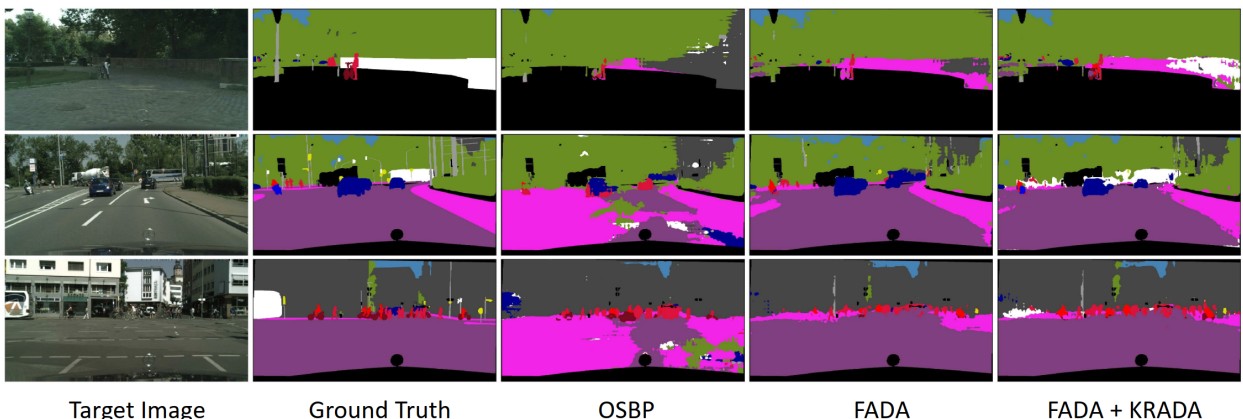

| Target Image | Ground Truth | OSBP | FADA | FADA + KRADA |

Figure 4: Qualitative results on SYNTHIA → Cityscapes. From left to right, each row is a target image, ground truth, and segmentation results of OSBP, FADA, and FADA + KRADA, where the white area denotes the unknown-class region.

Table 2 presents the results of GTA5 → Cityscapes. We can see that KRADA enables three CSDAS models to consistently outperform OSBP and improve the unknown-class IoU from 0% to nearly 1%, meanwhile obtaining margin gains in mIoU and mIoU* at the both last and best epochs. Overall, the results shown in Table 1 and Table 2 demonstrate that KRADA can enable CSDAS methods to identify unknown-class regions and promote a better overall adaptation by rejecting unknown-class regions and conducting the known-region-aware domain alignment. We also provide some qualitative segmentation results in Figure 4.

**Ablation studies:** 1) We tend to select the minority classes to construct the unknown class in the above experiments in order to retain as many source images as possible. Even though KRADA makes three CSDAS methods capable of detecting unknown regions, the unknown-class IoU values are not very large, especially for the GTA5 → Cityscape task in Table 2. This is because the unknown class is a challenging class to segment due to the imbalance issue. To explore the effect of different compositions of the unknown class, we conduct an extra ablation study. We choose car and light to form a new unknown class in GTA5 → Cityscape where the car is a majority class. The results are shown in Table 3. Compared to Table 2, there is a sharp overall decline in both mIoU and mIoU* since an easy class (car) is excluded from known classes. Comparatively, KRADA drastically increases the unknown-class IoU from 0% to more than 22% under the CLAN architecture and also improves mIoU and mIoU* by a large margin.

Table 3: Ablation study of varying the unknown class on GTA5 → Cityscapes.

| Method | | road | side. | build. | wall | pole | light | terrain | veg. | sky | person | rider | car | truck | bus | train | motor | bike | unk. | mIoU | mIoU* |
|---|---|---|---|---|---|---|---|---|---|---|---|---|---|---|---|---|---|---|---|---|---|
| CLAN (Luo et al., 2019) | B | 58.1 | 18.4 | 61.0 | 19.2 | 17.8 | 19.3 | 11.0 | 81.8 | 29.5 | 78.5 | 52.0 | 7.2 | 20.1 | 8.0 | 0.0 | 18.6 | 5.0 | 0.0 | 28.1 | 29.7 |
| | L | 53.8 | 17.8 | 59.9 | 17.9 | 16.5 | 16.2 | 8.2 | 82.2 | 29.4 | 79.4 | 51.3 | 4.3 | 19.2 | 6.7 | 0.0 | 15.6 | 3.3 | 0.0 | 26.8 | 28.3 |
| CLAN + KRADA | B | 64.1 | 12.9 | 71.5 | 19.9 | 19.2 | 15.9 | 10.6 | 81.2 | 31.0 | 77.1 | 50.3 | 7.0 | 21.4 | 10.9 | 0.0 | 20.3 | 8.1 | **25.3** | **30.4** | **30.7** |
| | L | 57.2 | 11.0 | 73.6 | 18.0 | 18.3 | 15.3 | 8.9 | 81.0 | 27.5 | 77.1 | 49.3 | 6.0 | 18.6 | 8.3 | 0.0 | 14.8 | 4.1 | **22.3** | **28.4** | **28.8** |

Table 4: Ablation study of the effects of the known-region map on SYNTHIA → Cityscapes. "w/o" indicates without the known-region map $R^{kn}$.

| Method | | road | side. | build. | fence | pole | sign | veg. | sky | person | rider | car | motor | bike | unk. | mIoU | mIoU* |
|---|---|---|---|---|---|---|---|---|---|---|---|---|---|---|---|---|---|
| CLAN + KRADA w/o $R^{kn}$ | B | 78.8 | 37.0 | 76.8 | 0.0 | 22.7 | 2.1 | 76.5 | 78.5 | 50.6 | 17.0 | 74.2 | 13.5 | 31.7 | 4.5 | 40.3 | 43.0 |
| | L | 72.8 | 37.4 | 76.3 | 0.0 | 21.7 | 2.2 | 75.5 | 79.2 | 50.7 | 16.3 | 72.7 | 13.4 | 29.2 | 3.9 | 39.4 | 42.1 |
| CLAN + KRADA | B | 82.4 | 37.3 | 76.4 | 0.0 | 22.1 | 2.5 | 76.6 | 77.8 | 49.9 | 18.5 | 72.4 | 15.3 | 28.9 | **5.3** | **40.4** | **43.1** |
| | L | 80.2 | 38.3 | 76.9 | 0.0 | 20.3 | 2.5 | 76.8 | 78.8 | 51.0 | 17.7 | 71.6 | 14.2 | 27.9 | **4.8** | **40.1** | **42.8** |

Table 5: Results of CLAN + KRADA with different statistical metrics on SYNTHIA → Cityscapes.

| Metrics | | road | side. | build. | fence | pole | sign | veg. | sky | person | rider | car | motor | bike | unk. | mIoU | mIoU* |
|---|---|---|---|---|---|---|---|---|---|---|---|---|---|---|---|---|---|
| MSP (Luo et al., 2020) | B | 81.9 | 37.6 | 77.8 | 0.0 | 19.4 | 2.2 | 77.8 | 79.6 | 50.3 | 18.2 | 61.4 | 14.7 | 34.3 | 5.1 | 40.0 | 42.7 |
| | L | 82.2 | 39.4 | 76.2 | 0.0 | 18.1 | 1.6 | 75.7 | 78.7 | 47.2 | 16.9 | 57.6 | 17.0 | 28.6 | 4.5 | 38.8 | 41.5 |
| L2 norm | B | 82.4 | 37.3 | 76.4 | 0.0 | 22.1 | 2.5 | 76.6 | 77.8 | 49.9 | 18.5 | 72.4 | 15.3 | 28.9 | **5.3** | 40.4 | 43.1 |
| | L | 80.2 | 38.3 | 76.9 | 0.0 | 20.3 | 2.5 | 76.8 | 78.8 | 51.0 | 17.7 | 71.6 | 14.2 | 27.9 | **4.8** | 40.1 | 42.8 |
| KL divergence | B | 80.2 | 37.4 | 77.6 | 0.1 | 23.2 | 2.8 | 77.0 | 78.5 | 49.4 | 17.7 | 80.4 | 15.0 | 30.1 | 3.9 | **40.9** | **43.8** |
| | L | 78.8 | 38.6 | 77.4 | 0.0 | 22.9 | 2.5 | 76.6 | 77.6 | 50.4 | 17.3 | 80.8 | 15.3 | 28.8 | 3.4 | **40.7** | **43.6** |

2) To investigate the impact of the known-region map $R^{kn}$, we compare the results of CLAN + KRADA with/without $R^{kn}$ on SYNTHIA → Cityscapes in Table 4. The performance of CLAN + KRADA w/o $R^{kn}$ is worse than CLAN + KRADA neither at the last epoch nor at the best epoch, which confirms the effectiveness of $R^{kn}$ and the necessity of rejection of unknown-class regions in target images for better domain alignment.

3) We also compare the results of three statistical metrics (MSP, L2 norm, and KL divergence) used for generating unknown-class pseudo labels in Table 5. MSP achieves satisfactory and comparable results and L2 norm achieves the highest unknown-class IoU. Although KL divergence obtains relatively lower unknown-class IoU, it has superior performance in segmenting known classes (mIoU*) and the best segmentation overall performance (mIoU). Both L2 norm and KL divergence leverage the property of the unknown class to compare the output probability distribution of a pixel in the test image with the known-class prior probability distribution. This indicates that the statistical metric is not fixed and also proves the adjustability and flexibility of KRADA. However, MSP does not utilize the property of the unknown class derived from the source data, which may be the main reason why MSP achieves a slightly inferior performance.

4) To extend our proposed method in more segmentation settings, we have conducted experiments to prove that this method can be applied to the same domain, i.e., only with unknown class shifts without target domain shifts. Therefore, we use the abandoned SYNTHIA images (those with the unknown class) as the target domain and show the results in Table 6. The results show that the proposed method not only can be applied to OSDAS (with unknown class shifts and target domain shifts) but also is applicable to the same-domain setting (with only unknown class shifts).

5) To evaluate the effectiveness of using the known-class prior distribution to depict the distribution characteristics of the unknown class, we also use the uniform distribution as a baseline and make a comparison in Table 7. Although the uniform distribution achieves similar mIoU and mIoU* as the known-class prior distribution at the best epoch, its unknown-class IoU values are significantly lower at the both last and best epochs. Table 7 shows that the known-class prior distribution is more suitable to portray the distribution characteristics of the unknown class while taking full advantage of the data attribute of source data.

Table 6: Results of CLAN + KRADA applied to the same domain (SYNTHIA → SYNTHIA with the unknown class).

| Setting | | road | side. | build. | fence | pole | sign | veg. | sky | person | rider | car | motor | bike | unk. | mIoU | mIoU* |
|---|---|---|---|---|---|---|---|---|---|---|---|---|---|---|---|---|---|
| Same domain | B | 75.9 | 44.5 | 80.0 | 26.6 | 39.72 | 0.2 | 60.3 | 81.3 | 54.3 | 41.6 | 60.6 | 23.7 | 28.6 | 8.6 | 44.7 | 47.5 |
| | L | 69.7 | 44.4 | 85.0 | 27.0 | 39.5 | 0.1 | 63.0 | 81.8 | 56.1 | 36.7 | 57.5 | 17.6 | 24.8 | 7.7 | 43.6 | 46.4 |
| OSDAS | B | 82.4 | 37.3 | 76.4 | 0.0 | 22.1 | 2.5 | 76.6 | 77.8 | 49.9 | 18.5 | 72.4 | 15.3 | 28.9 | 5.3 | 40.4 | 43.1 |
| | L | 80.2 | 38.3 | 76.9 | 0.0 | 20.3 | 2.5 | 76.8 | 78.8 | 51.0 | 17.7 | 71.6 | 14.2 | 27.9 | 4.8 | 40.1 | 42.8 |

Table 7: Results of CLAN + KRADA with different $p^{prior}$ for the unknown-class pseudo-label generation on SYNTHIA → Cityscapes.

| Uknown-class | | road | side. | build. | fence | pole | sign | veg. | sky | person | rider | car | motor | bike | unk. | mIoU | mIoU* |
|---|---|---|---|---|---|---|---|---|---|---|---|---|---|---|---|---|---|
| Uniform distribution | B | 80.6 | 36.8 | 77.8 | 0.0 | 24.6 | 2.4 | 76.7 | 77.1 | 49.3 | 18.1 | 81.7 | 14.3 | 30.7 | 2.1 | **40.9** | **43.9** |
| | L | 77.6 | 37.4 | 77.0 | 0.0 | 22.4 | 2.0 | 76.2 | 75.2 | 48.5 | 17.3 | 80.1 | 13.8 | 29.8 | 1.6 | 39.9 | 42.9 |
| Known-class prior distribution | B | 80.2 | 37.4 | 77.6 | 0.1 | 23.2 | 2.8 | 77.0 | 78.5 | 49.4 | 17.7 | 80.4 | 15.0 | 30.1 | **3.9** | **40.9** | 43.8 |
| | L | 78.8 | 38.6 | 77.4 | 0.0 | 22.9 | 2.5 | 76.6 | 77.6 | 50.4 | 17.3 | 80.8 | 15.3 | 28.8 | **3.4** | **40.7** | **43.6** |

Table 8: Results of CLAN + KRADA with single and two classification heads on SYNTHIA → Cityscapes.

| Classification heads | | road | side. | build. | fence | pole | sign | veg. | sky | person | rider | car | motor | bike | unk. | mIoU | mIoU* |
|---|---|---|---|---|---|---|---|---|---|---|---|---|---|---|---|---|---|
| C | B | 67.3 | 36.0 | 76.1 | 0.0 | 20.6 | 2.1 | 72.1 | 73.5 | 48.5 | 16.3 | 64.9 | 11.9 | 28.3 | 3.5 | 37.2 | 39.8 |
| | L | 55.7 | 31.3 | 75.5 | 0.0 | 18.3 | 1.8 | 75.1 | 76.8 | 48.7 | 17.0 | 62.7 | 13.7 | 30.0 | 2.9 | 36.4 | 39.0 |
| C and C* | B | 82.4 | 37.3 | 76.4 | 0.0 | 22.1 | 2.5 | 76.6 | 77.8 | 49.9 | 18.5 | 72.4 | 15.3 | 28.9 | **5.3** | 40.4 | **43.1** |
| | L | 80.2 | 38.3 | 76.9 | 0.0 | 20.3 | 2.5 | 76.8 | 78.8 | 51.0 | 17.7 | 71.6 | 14.2 | 27.9 | **4.8** | 40.1 | **42.8** |

Table 9: Results of CLAN + KRADA with varying $\gamma$ on SYNTHIA → Cityscapes.

| $\gamma$ | | road | side. | build. | fence | pole | sign | veg. | sky | person | rider | car | motor | bike | unk. | mIoU | mIoU* |
|---|---|---|---|---|---|---|---|---|---|---|---|---|---|---|---|---|---|
| 0.005% | B | 79.8 | 38.2 | 77.3 | 0.0 | 22.1 | 2.3 | 76.9 | 81.0 | 51.9 | 19.3 | 78.8 | 20.7 | 28.1 | 0.1 | **41.2** | **44.4** |
| | L | 82.1 | 37.5 | 77.8 | 0.0 | 24.4 | 2.5 | 78.0 | 78.4 | 46.5 | 18.8 | 80.9 | 13.1 | 28.2 | 0.1 | **40.6** | **43.7** |
| 0.01% | B | 80.3 | 39.4 | 78.2 | 0.1 | 24.8 | 2.8 | 78.0 | 79.3 | 49.4 | 18.0 | 80.8 | 12.9 | 31.0 | 1.6 | **41.2** | 44.2 |
| | L | 81.0 | 38.6 | 78.0 | 0.0 | 24.0 | 2.5 | 78.0 | 76.8 | 47.7 | 17.5 | 79.6 | 11.3 | 29.8 | 1.0 | 40.4 | 43.4 |
| **0.05%** | B | 82.4 | 37.3 | 76.4 | 0.0 | 22.1 | 2.5 | 76.6 | 77.8 | 49.9 | 18.5 | 72.4 | 15.3 | 28.9 | **5.3** | 40.4 | 43.1 |
| | L | 80.2 | 38.3 | 76.9 | 0.0 | 20.3 | 2.5 | 76.8 | 78.8 | 51.0 | 17.7 | 71.6 | 14.2 | 27.9 | **4.8** | 40.1 | 42.8 |
| 0.1% | B | 79.0 | 35.8 | 76.6 | 0.0 | 21.4 | 1.8 | 76.5 | 76.5 | 47.4 | 17.7 | 72.9 | 14.2 | 27.2 | 4.6 | 39.4 | 42.1 |
| | L | 73.9 | 32.0 | 73.9 | 0.0 | 19.3 | 1.9 | 74.4 | 77.0 | 47.5 | 16.2 | 61.2 | 11.8 | 24.9 | 4.0 | 37.0 | 39.5 |
| 0.5% | B | 70.3 | 31.9 | 73.7 | 0.1 | 19.7 | 0.9 | 69.8 | 71.9 | 48.9 | 18.3 | 64.4 | 16.5 | 33.6 | 3.9 | 37.4 | 40.0 |
| | L | 49.9 | 11.3 | 48.0 | 0.0 | 10.2 | 0.1 | 34.6 | 45.4 | 27.1 | 7.5 | 28.7 | 6.0 | 8.5 | 2.1 | 20.0 | 21.3 |

6) To verify the necessity of two classification heads C and C*, we conduct the ablation study in Table 8. Experimental results demonstrate that two classification heads are necessary since these two classification heads have different functions and do not mix with each other.

7) To show the effects of different $\gamma$, we change the value of $\gamma$ and show the results in Table 9. Compared with 0.05%, reducing $\gamma$ (0.005% and 0.01%) leads to the decrease of identifying the unknown-class pixels (lower unknown-class IoU) and the increase of segmenting known classes and overall performance (mIoU* and mIoU). On the contrary, increasing $\gamma$ (0.1% and 0.5%) does not mean a higher unknown-class IoU and instead, it diminishes the known-class segmentation performance and thus an inferior overall performance (lower mIoU* and mIoU). More specifically, we achieve unsatisfactory and unstable results when $\gamma$ is set as 0.5%. To obtain satisfactory and competitive segmentation performance, it is appropriate to set $\gamma$ between 0.01% – 0.1%. Therefore, we finally set $\gamma$ as 0.05%.

8) Comparison with other related works. Since there is no relevant work to solve the scenario that is completely consistent with ours: open set, domain adaptation, and semantic segmentation, other methods are not easy and suitable to directly be compared with our proposed method. Thus we adopt a compromise way to find the comparable part that is unknown-class pseudo-label generation criteria. Therefore, we compared the

Table 10: Comparison with other related works.

| Metrics | | road | side. | build. | fence | pole | sign | veg. | sky | person | rider | car | motor | bike | unk. | mIoU | mIoU* |
|---|---|---|---|---|---|---|---|---|---|---|---|---|---|---|---|---|---|
| MSP (Luo et al., 2020) | B | 81.9 | 37.6 | 77.8 | 0.0 | 19.4 | 2.2 | 77.8 | 79.6 | 50.3 | 18.2 | 61.4 | 14.7 | 34.3 | 5.1 | 40.0 | 42.7 |
| | L | 82.2 | 39.4 | 76.2 | 0.0 | 18.1 | 1.6 | 75.7 | 78.7 | 47.2 | 16.9 | 57.6 | 17.0 | 28.6 | 4.5 | 38.8 | 41.5 |
| DML (Cen et al., 2021) | B | 77.3 | 37.4 | 77.3 | 0.0 | 23.9 | 1.6 | 75.9 | 68.6 | 48.4 | 16.5 | 76.7 | 12.8 | 30.7 | 0.9 | 39.1 | 42.1 |
| | L | 72.6 | 35.9 | 77.3 | 0.0 | 22.0 | 1.7 | 71.4 | 62.7 | 47.3 | 18.5 | 79.8 | 9.7 | 37.3 | 1.4 | 38.4 | 41.3 |
| MSP-baseline (Vaze et al., 2021) | B | 82.5 | 35.1 | 75.0 | 0.0 | 19.7 | 2.4 | 74.7 | 77.2 | 47.6 | 19.3 | 78.9 | 7.0 | 29.2 | 2.5 | 39.4 | 42.2 |
| | L | 79.5 | 35.6 | 73.6 | 0.0 | 17.4 | 2.1 | 74.1 | 76.3 | 48.4 | 19.0 | 73.8 | 7.5 | 29.7 | 2.2 | 38.5 | 41.3 |
| L2 norm | B | 82.4 | 37.3 | 76.4 | 0.0 | 22.1 | 2.5 | 76.6 | 77.8 | 49.9 | 18.5 | 72.4 | 15.3 | 28.9 | **5.3** | 40.4 | 43.1 |
| | L | 80.2 | 38.3 | 76.9 | 0.0 | 20.3 | 2.5 | 76.8 | 78.8 | 51.0 | 17.7 | 71.6 | 14.2 | 27.9 | **4.8** | 40.1 | 42.8 |
| KL divergence | B | 80.2 | 37.4 | 77.6 | 0.1 | 23.2 | 2.8 | 77.0 | 78.5 | 49.4 | 17.7 | 80.4 | 15.0 | 30.1 | 3.9 | **40.9** | **43.8** |
| | L | 78.8 | 38.6 | 77.4 | 0.0 | 22.9 | 2.5 | 76.6 | 77.6 | 50.4 | 17.3 | 80.8 | 15.3 | 28.8 | 3.4 | **40.7** | **43.6** |

Table 11: Results on COVID-19 infection segmentation.

| Method | Lung_IoU | Infection_IoU | Accuracy | Precision | Recall | F1-score |
|---|---|---|---|---|---|---|
| OSBP (Saito et al., 2018b) | 69.9 | 0.6 | 66.7 | 66.7 | 100.0 | 80.0 |
| AdaptSegNet + KRADA | 80.7 | 0.4 | 66.7 | 66.7 | 100.0 | 80.0 |
| CLAN + KRADA | 81.7 | 0.8 | 66.7 | 66.7 | 100.0 | 80.0 |
| FADA + KRADA | **85.6** | **2.0** | **100.0** | **100.0** | **100.0** | **100.0** |

performance of different unknown-class pseudo-label generation criteria: including the MSP in (Luo et al., 2020) and DML in (Cen et al., 2021). We realized MSP and DML in CLAN + KRADA framework and showed the results of SYNTHIA → Cityscapes in Table 10. In addition, Vaze *et al.* (Vaze et al., 2021) introduced a baseline for open-set recognition (OSR). The OSR baseline refers to training a $K$-way classifier and identifying the unknown pixels directly during inference by assigning the unknown-class probability with 1 minus the maximum value of softmax probabilities. We denoted this OSR baseline as MSP-baseline to distinguish it from MSP (Luo et al., 2020). Table 10 also shows the results of MSP-baseline on CLAN for the SYNTHIA → Cityscapes task. Our proposed methods significantly outperform MSP-baseline and DML with large margins in terms of unknown-class IoU, mIoU, and mIoU*.

**Real-world OSDAS task (COVID-19 infection segmentation in CT scans).** To construct a COVID-19 task, we exploit the public datasets summarized in (Ma et al., 2020). The source data consists of normal CT scans with lung annotations. Both target data and test data include COVID-19 cases and non-infected CT scans. Detailed data descriptions and data processes are introduced in Appendix B. To give a comprehensive comparison, we evaluate the proposed method from both pixel-level and instance-level aspects. The IoU values of lung and infection averaged among all test cases are reported in Table 11, denoted as Lung_IoU and Infection_IoU. We also provide four other metrics: Accuracy, Precision, Recall, and F1-score, commonly used in medical fields for instance-level evaluation.

**Results on real-world OSDAS task.** In Table 11, all these models achieve 100.0% recall, meaning that all infected cases are detected, which is desirable in medical image diagnosis. OSBP gains a 0.6% IoU in infection and the same instance-level performance as AdaptSegNet + KRADA and CLAN + KRADA, but it heavily sacrifices the segmentation accuracy of the lung. Compared with OSBP, CLAN + KRADA shows superior performance in both pixel-level and instance-level evaluations. Moreover, FADA + KRADA greatly outperforms other models and obtains the best results. Examples of segmentation results are provided in Figure 5.

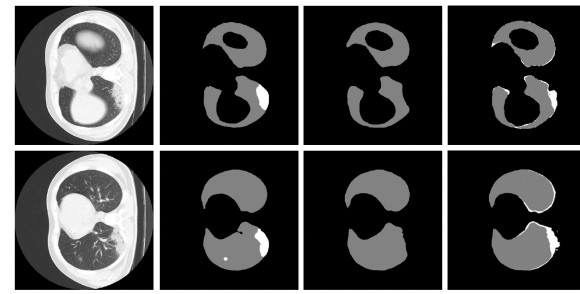

Figure 5: From left to right: CT slice, Ground Truth, and results of FADA and FADA + KRADA, where the brighter area is the infected area.

## 7 Conclusion

This paper considers the semantic segmentation task in an open world, where test images have a different distribution from training images and contain unknown categories/classes. To address this new and challenging problem, we explore the inherent property of unknown classes and propose an end-to-end framework, KRADA, that performs known-region-aware domain alignment. KRADA is a generalized framework with no particular structure dependency and can be easily implemented on existing CSDAS methods, as demonstrated by three realizations of KRADA in our experiments. Experimental results validate that KRADA enables CSDAS methods to distinguish unknown-class pixels from known-class pixels and classify known-class pixels well.

**Acknowledgments**

CHZ and BH were supported by NSFC Young Scientists Fund No. 62006202, Guangdong Basic and Applied Basic Research Foundation No. 2022A1515011652, RGC Early Career Scheme No. 22200720, RGC Research Matching Grant Scheme No. RMGS20221102, No. RMGS20221306 and No. RMGS20221309. BH was also supported by HKBU CSD Departmental Incentive Grant. CG is supported by NSF of China (No: 61973162), NSF of Jiangsu Province (No: BZ2021013), NSF for Distinguished Young Scholar of Jiangsu Province (No: BK20220080), and CAAI-Huawei MindSpore Open Fund. TLL was partially supported by Australian Research Council Projects IC-190100031, LP-220100527, DP-220102121, and FT-220100318. RFZ was partially supported by NSF of China No. 62172083. This research is partially supported by the Research Matching Grant Scheme RMGS2021_8_06 from the Hong Kong Government.

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

# A   Related Works

In this section, we briefly review two kinds of domain adaptation settings related to open-set domain adaptation segmentation (OSDAS).

**Unsupervised open set domain adaptation (UOSDA)** is first proposed by Busto *et al.* (Panareda Busto & Gall, 2017), where both source and target domain contain private and shared classes, but we only know shared labels. To relax the requirement of private source labels, Saito *et al.* (Saito et al., 2018b) introduce a new concept of UOSDA setting where only the target domain has private labels and propose Open Set Back-Propagation (OSBP), a novel adversarial training-based method for an open set scenario. Later on, UOSDA methods follow this new and realistic setting and cope with it by a coarse-to-fine weighting mechanism (Liu et al., 2019), a self-supervised task (Bucci et al., 2020), self-ensembling (Pan et al., 2020), or inheritable models (Kundu et al., 2020). Additionally, Fang *et al.* and Zhong *et al.* (Fang et al., 2020; Zhong et al., 2020) provide theoretical analysis for UOSDA and introduce open set difference, a special term that facilitates recognizing unknown target samples. Feng *et al.* (Feng et al., 2019) explicitly utilize the semantic margin of open set data to make the unknown class apart from the decision boundary and the known classes more separable. Luo *et al.* (Luo et al., 2020) propose a graph neural network with episodic training and achieve state-of-the-art performance. However, these existing UOSDA methods only focus on classification tasks, and most of them cannot be modified for semantic segmentation by simply convolutionalizing their classification architectures. For example, the state-of-the-art UOSDA method (Luo et al., 2020) requires source episodes containing each known class in a batch to construct a graph neural network. This requirement is hard to meet in segmentation tasks which demand a large memory to support dense computations. Therefore, modifying existing UOSDA methods for OSDAS is not usually feasible.

**Closed set domain adaptation for semantic segmentation (CSDAS)** has been extensively studied and developed maturely (Toldo et al., 2020). A predominant stream of UCSDA works is adversarial training (AT) based methods which minimize adversarial losses to align the distributions between the source and target domains at input level (Hoffman et al., 2018; Gong et al., 2019), feature level (Hoffman et al., 2016; Vu et al., 2019; Chen et al., 2019; Wang et al., 2020; Hoffman et al., 2018), or output space level (Luo et al., 2019; Saito et al., 2018a; Tsai et al., 2018). Recently, a series of approaches (Zou et al., 2018; Mei et al., 2020; Zou et al., 2019; Zhang et al., 2019) based on deep self-training (ST) has become an alternative research direction. These approaches generate pseudo-labels for target samples to provide extra supervision so that the network can be trained under the supervision of two domains. Zou *et al.* (Zou et al., 2018) propose a class-balanced self-training (CBST) framework to overcome the issue of imbalanced target pseudo-labels. To avoid overconfident wrong pseudo-labels, Zou *et al.* (Zou et al., 2019) further incorporate two types of confidence regularization to CBST. To generate high-quality pseudo-labels, Mei *et al.* (Mei et al., 2020) propose an instance adaptive selector to generate more accurate pseudo-labels. In addition, several works (Li et al., 2019; Tsai et al., 2019) have been developed by combining AT and ST methods and presented great potential. Despite the well-studied CSDAS methods, they cannot detect unknown classes and lead to negative transfer due to mismatched label sets. Therefore, current CSDAS methods are not applicable to an open world and cannot solve OSDAS tasks well.

Based on such well-studied CSDAS methods, we propose an end-to-end framework, KRADA, which can modify current CSDAS segmentation methods and adapt them for OSDAS tasks.

# B   Experiment Setup

## B.1   Experiments on Two Synthetic OSDAS Tasks

**Data description:** For the task SYNTHIA → Cityscapes, SYNTHIA originally includes 9,400 synthetic images and has 16 common classes with Cityscapes. These three classes (wall, light, and bus) are selected to form the unknown class, and we discard those images containing either of the three classes. The remaining data in SYNTHIA contain 750 images, and there are 13 shared known classes in this task. For the task GTA5 → Cityscapes, GTA5 initially contains 24,966 images rendered from the GTA5 game engine and has the same 19 category annotations as Cityscapes. Similarly, we choose two classes (fence and sign) as the unknown

class and only retain the images which do not contain the unknown class. The remaining images in GTA5 include 17 categories and 2,277 images. Cityscapes is a real-world dataset consisting of a training set with 2,957 images and a validation set with 500 images. We divide the Cityscapes training set into training splits of 2,500 images for training and evaluation splits of 457 images for hyperparameter selection. The results on the Cityscapes validation set are reported for performance comparison.

**Implementation details:** For a fair comparison, we adopt the Deeplab-V2 (Chen et al., 2017) framework with ResNet-101 (He et al., 2016) pretrained on ImageNet (Deng et al., 2009) as the segmentation base network. We implement KRADA on three CSDAS methods: AdaptSegNet (Tsai et al., 2018), CLAN (Luo et al., 2019), and FADA (Wang et al., 2020). For each CSDAS method, we additionally duplicate a last convolution classification module as $C^*$ which is arranged in parallel with the original classifier $C$ after the feature extractor. $C^*$ is identical as $C$ expect for the last layer with channel number $K$ to output the predicted score map over known classes. Regarding the pseudo-label hyperparameters in KRADA, $\gamma$, $\beta$, and $\delta$ are chosen as 0.05%, 0.99, and 0.1 respectively for these two synthetic segmentation tasks. $\alpha$ is set as 0.1, 0.03, and 0.2 for the AdaptSegNet, CLAN, and FADA models equipped with KRADA. Other experimental settings such as discriminator structure, optimization policy, and hyperparameters in three modified models are almost the same as those described in the original papers. All the models are implemented using Python 3.6 and Pytorch 1.7 on a TITAN Tesla V100 GPU.

## B.2 Experiments on COVID-19 Infection Segmentation in CT Scans

We describe the details of our data used in the COVID-19 infection segmentation task. More specifically, we exploit the public datasets summarized in (Ma et al., 2020) to construct a real-world OSDAS task. Source data includes 30 CT scans which are randomly selected from NSCLC left and right lung segmentation dataset (Kiser et al., 2020; Aerts et al., 2014; Clark et al., 2013) (with CC BY-NC license). The target data consists of 20 COVID-19 CT scans and 10 non-infected CT scans. More specifically, COVID-19 CT scans are publicly available (Jun et al., 2020) (with CC BY-NC-SA license), and each case has the annotations of the left lung, right lung, and infection. Non-infected CT scans are randomly selected from MICCAI 2019 StructSeg lung organ segmentation challenge, and we only use their left lung and right lung annotations. We divide the target data into two parts for training and testing, and each part includes 10 COVID-19 cases and 5 non-infected CT scans. Following (Ma et al., 2020), we adjust each CT scan to lung window [-1250, 250] and then normalize it to [0,255] for pre-processing. We slice each CT volume into 2D slices and perform the same data augmentation as (Müller et al., 2020). In this task, the pseudo-label hyperparameters $\gamma$, $\beta$, and $\delta$ are chosen as 1%, 0.99, and 0.001 respectively. $\alpha$ is set as 0.01, 0.01, and 0.1 for the three models equipped with KRADA. Other experimental setups are similar to the above two synthetic tasks.

## C Potential Social Impacts

Our research allows training a segmentation model for a new dataset by exploiting existing annotated data, which reduces the annotation cost. Besides, the proposed method can recognize the abnormal and unseen regions in a new dataset and give an early warning. This is quite critical and has a broader significance in the medical field, especially when it comes to a new disease, and we know nothing about it. However, this method is not entirely mature, and there are potential risks of false alarms and missed detection. Therefore, it cannot be used in clinical medical practice for the time being.

