# OpenReview forum: "KRADA: Known-region-aware Domain Alignment for Open-set Domain Adaptation in Semantic Segmentation"
_TMLR — Accepted by TMLR_

### Review · Reviewer_r9Vn · 2022-12-22

**Summary Of Contributions:**

This paper considers the problem of open-world semantic segmentation, where the unlabeled data include both domain shift and category shift. To solve this problem, this authors first propose an unknown-class estimation approach and design a  known-region-aware domain alignment framework to carefully align the data distribution of source and target domains. Experiments conducted on two tasks show the benefit of the proposed method.

**Audience:**

No

**Claims And Evidence:**

Yes

**Requested Changes:**

Please solve my concerns in the weaknesses.

- Address the novelty of the proposed method.

- Make the motivation more clear.

- Explain the large improvement of the proposed method.

- Better verify the effectiveness of unknown-class pseudo-label generation.

- State the rationality of COVID-19 dataset under the proposed open-world setting.

- Compare with more recent open-set domain adaptation methods in the community.

**Strengths And Weaknesses:**

*Strengths*

+ This paper considers a new, rarely studied problem in semantic segmentation. It is an interesting and important task in the community.

+ An unknown-class estimation is proposed to detect unknown pixels in the target domain.

+ A known-region-aware domain alignment framework is introduced to perform a safe domain alignment under the open-world setting.


*Weaknesses*

- Although this paper solves a new setting, the novelty of the proposed method is somewhat limited. I am not supervised by the unknown-class estimation and the known-region-aware domain alignment methods. At least, I do not well agree these two techniques can meet the requirement of TMLR.

- The motivation of unknown-class pseudo-label generation is not well-explained. Is there any theoretical or experimental supports?

- The improvement of the proposed method is not consistent. For example, in many cases, the improvement of the proposed method is limited.

- Except the final segmentation results, the authors should provide other metrics the verify the benefit of the proposed unknown-class pseudo-label generation. For example, the accuracy of the selected pseudo-labels.

- For the COVID-19 dataset, do we need to ignore the new-infected region (unknown class) or detect? I think it is important to detect the new-infected region. However, in the proposed open-world setting and method, it seems the unknown-region should not be considered during training and detected during a testing, which I think is not reasonable in practice.

- Many recent open-set domain adaptation methods are not considered.

---

> ### Author Response · Authors · 2023-01-16
> **Response to Reviewer r9Vn (Part 1)**
>
> Thank you for your detailed review. we hope the following responses could address your concerns:
>
> #### __Q1: About the novelty.__
>  __A1:__ In this work, we aim to propose a simple and general approach based on the well-studied closed set domain adaptation for semantic segmentation (CSDAS) methods, so that we can make as few modifications as possible to adapt CSDAS methods for a new OSS task. What we pursue is that the approach is simple and practical. It can leverage the existing CSDAS methods and modify them slightly to handle the open world setting.
>
> Besides a novel proposed concept: the known-region map, it is also worth emphasizing that we explicitly point out the distribution characteristics of unknown classes, considering the prior knowledge of source data. Thanks to this characteristic, we can utilize a statistical metric to describe how well the output softmax distribution of a target pixel ﬁts the distribution of an unknown class, which is different from the common practice to generate unknown-class pseudo-labels by comparing the maximum softmax probability (MSP) or entropy with a threshold [r1]. MSP or entropy only uses the softmax output and does not consider the essential distribution characteristic of unknown classes, which merely depicts the internal relationships of known classes and does not measure the distance of a pixel to an unknown class in a strict sense. (We also emphasized the difference in Remark 3 in the manuscript). In addition, there is less research to explore what the unknown class and its essential characteristics are. We propose the known-class prior distribution to depict the distribution characteristic of the unknown class, which has been proven to be rational and feasible. We hope our work can give the inspiration to model the unknown class.
>
> Furthermore, the statistical metric used to measure the distribution disparity is not unique and can be replaced and the effectiveness of KRADA has been verified on three different CSDAS methods, which shows the adjustability and ﬂexibility of the proposed method.
>
>
> #### __Q2: About the motivation of unknown-class pseudo-label generation.__
>  __A2:__ In the closed set domain adaptation for semantic segmentation (CSDAS) setting, it is a common practice to introduce the known-class pseudo-labels by selecting the pixels with high confidence/maximum softmax probability. Similarly, we aim to introduce the unknown-class pseudo-labels in the open set domain adaptation for semantic segmentation (OSS). As for the unknown-class pseudo-labels, a common practice is to compare the maximum softmax probability or entropy with a threshold and mark the data with high uncertainty as an unknown class [r1].  In principle, these researchers regard prediction uncertainty as a criterion to determine whether a new input is from the unknown class. Nevertheless, it should be noted that uncertainty is not unknown [r3], and thresholding on uncertainty is not sufﬁcient to determine what is unknown [r4].
>
> Therefore, different from those which rely on a criterion to select uncertain pixels as unknown pixels, our starting point is to explore what is unknown and characterize the unknown classes as far as possible. As we all know, it is hard to accurately depict the unknown classes, due to the complexity and diversity of “unknown” classes in the real world. A plain and ideal condition that an unknown class is independent of known classes, and its prediction probabilities could not bias any known classes ideally, i.e., its distribution conforms to a uniform distribution. But this assumption is very strong and ideal and means that we have no any prior knowledge about the target domain. Benefiting from the prior knowledge of source data, we can relax this assumption by utilizing the class prior probability of source classes. Experimental results verify that this assumption about the distribution characteristic of the unknown class is rational and feasible.

---

> > ### Author Response · Authors · 2023-01-16
> > **Response to Reviewer r9Vn (Part 2)**
> >
> > #### __Q3: Explain the large improvement of the proposed method.__
> >  __A3:__ Table 1 shows the effectiveness of KRADA on three CSDAS methods for SYNTHIA → Cityscapes task. KRADA consistently improves the unknown-class IoU from 0 to about 5% and above meanwhile improving the known-class segmentation and the overall segmentation performance (higher mIoU* and mIoU, compared with the original three CSDAS methods).
> >
> > Similarly, Table 2 shows the effectiveness of KRADA for GTA5 → Cityscapes task, but the unknown-class IoU is no more than 1%, which maybe the main reason why the reviewer thinks the limited improvement of the proposed method. We also discussed this phenomenon and conduct the ablation studies in page 9. For the GTA5 → Cityscapes task, we choose two classes (fence and sign) as the unknown class and only retain the source images which do not contain the unknown class. Fence and sign are minority classes, so we can retain as many source images as possible to train the model. Due to the rare number of unknown-class pixels, this segmentation task becomes more challenging, which leads to unknown-class IoU values in Table 2 not being very large. This explanation is also applicable to the COVID task. To eliminate the impacts of the constructed unknown class, we change the different composition of the unknown class (chose car and light to form a new unknown class and car is a majority class) and show the results in Table 3. KRADA drastically increases the unknown-class IoU from 0% to more than 22%.
> >
> > #### __Q4: About the effectiveness of unknown-class pseudo-label generation.__
> >  __A4:__  We evaluate the quality of generated unknown-class pseudo-labels from two aspects: pixel-level and instance-level. The unknown-class IoU is used for the pixel-level evaluation and Accuracy, Precision, Recall, and F1-score for instance-level evaluation. The results in SYNTHIA → Cityscapes task are:
> > Unknown-class IoU:  0.61%;
> > Accuracy, Precision, Recall, and F1-score: 71.68%, 71.79%, 99.78%, 83.50%. The actual unknown-class pixels.
> > Considering the actual unknown class pixels account for a small proportion: 0.97%, the quality of generated unknown-class pixels is acceptable.
> >
> > #### __Q5: About the rationality of COVID-19 dataset under the proposed open-world setting.__
> >  __A5:__ For the COVID-19 infection segmentation task, we used the non-infected normal CT scans with lung annotations as the source data, and these source data are collected before COVID-19 occurs. Particularly, there is no knowledge about this disease at the beginning of COVID-19 outbreaks while we have a lot of non-infected normal CT scans to utilize. This is a typical OSS task scenario. If the unknown-regions are seen in the training phase, this scenario becomes a closed set domain adaptation for semantic segmentation (CSDAS) task, which can be easily solved by well-studied CSDAS methods. Our proposed method can extend the existing CSDAS methods and modify them slightly to handle the open world setting.
> >
> > #### __Q6: Compare with more recent open-set domain adaptation methods in the community.__
> >  __A6:__ In fact, there is no relevant work to solve the scenario that is completely consistent with ours: open set, domain adaptation, and semantic segmentation. Therefore, we review two kinds of much more relevant related works: 1) unsupervised open set domain adaptation (UOSDA) classification and 2) closed set domain adaptation for semantic segmentation (CSDAS) in the supplementary material. We reviewed a group of works about UOSDA and explicitly pointed out that these existing UOSDA methods only focus on classification tasks and most of them are not easily modified for semantic segmentation. Fortunately, Open Set Back Propagation (OSBP) proposed by Saito et al. [r5] can be applied to semantic segmentation by convolutionalizing its classification architecture.  Besides, the new setting of UOSDA is also proposed by Saito et al. [r5]. Therefore, we choose to compare our proposed method with OSBP (seen in Table 1, 2 and 5), which is representative.
> >
> > We recently found that Cen et al. solves a similar open-set semantic segmentation setting in [r7], but they did not care about the distribution shift between source data and target data. In addition, Cen et al. adopted a few-shot learning method which needs labels to give additional annotations. Their work is very different from ours and we have to try to find the comparable part that is unknown-class pseudo-label generation method.
> >
> > Therefore, we compared the performance of different unknown-class pseudo-label generation criteria: including the MSP in Vaze et al. [r6], DML in Cen et al. [r7], and show the results in the bellow table:

---

> > > ### Author Response · Authors · 2023-01-16
> > > **Response to Reviewer r9Vn (Part 3)**
> > >
> > > __Table  10: Comparison with other related works.__
> > > | Metrics             |      | road | side. | build.| fence | pole | sign | veg. | sky | person | rider | car | motor | bike | unk. | mIoU          | mIoU* |
> > > |-------------------------|------|-----------------|-----------------|--------------------|-----------------------|-------------|----------------------|----------------------|---------------------|--------------|--------------|-----------------|----------------|--------------|---------------|----------|-----------|
> > > | MSP (Luo et al.) | B    | 81.9                 | 37.6                  | 77.8                   | 0.0                   | 19.4                 | 2.2                  | 77.8                 | 79.6                | 50.3                   | 18.2                  | 61.4                | 14.7                  | 34.3                 | 5.1                  | 40.0          | 42.7          |
> > > |                                         | L    | 82.2                 | 39.4                  | 76.2                   | 0.0                   | 18.1                 | 1.6                  | 75.7                 | 78.7                | 47.2                   | 16.9                  | 57.6                | 17.0                  | 28.6                 | 4.5                  | 38.8          | 41.5          |
> > > | DML (Cen et al.) | B    | 77.3                 | 37.4                  | 77.3                   | 0.0                   | 23.9                 | 1.6                  | 75.9                 | 68.6                | 48.4  | 16.5                 | 76.7                  | 12.8                   | 30.7                  | 0.9                  | 39.1                 | 42.1                 |
> > > |                                         | L    | 72.6                 | 35.9                  | 77.3                   | 0.0                   | 22.0                 | 1.7                  | 71.4                 | 62.7                | 47.3                   | 18.5                  | 79.8                | 9.7                   | 37.3                 | 1.4                  | 38.4          | 41.3          |
> > > | MSP* (Vaze et al.) | B    | 82.5                 | 35.1                    | 75.0                      | 0.0                   | 19.7                 | 2.4                  | 74.7               | 77.2                | 47.6                  | 19.3                  | 78.9                | 7.0                  | 29.2                 | 2.5                  | 39.4          | 42.2          |
> > > |                                         | L    | 79.5                 | 35.6                     | 73.6                     | 0.0                   | 17.4                 | 2.1                  | 74.1                 | 76.3                | 48.4                   | 19.0                  | 73.8                | 7.5                  | 29.7                 |2.2                  | 38.5          | 41.3         |
> > > | L2 norm                  | B    | 82.4                 | 37.3                  | 76.4                   | 0.0                   | 22.1                 | 2.5                  | 76.6                 | 77.8                | 49.9                   | 18.5                  | 72.4                | 15.3                  | 28.9                 | **5.3**         | 40.4          | 43.1          |
> > > |                                         | L    | 80.2                 | 38.3                  | 76.9                   | 0.0                   | 20.3                 | 2.5                  | 76.8                 | 78.8                | 51.0                   | 17.7                  | 71.6                | 14.2                  | 27.9                 | **4.8**         | 40.1          | 42.8          |
> > > | KL divergence            | B    | 80.2                 | 37.4                  | 77.6                   | 0.1                   | 23.2                 | 2.8                  | 77.0                 | 78.5                | 49.4                   | 17.7                  | 80.4                | 15.0                  | 30.1                 | 3.9                  | **40.9** | 	**43.8** |
> > > |                                         | L    | 78.8                 | 38.6                  | 77.4                   | 0.0                   | 22.9                 | 2.5                  | 76.6                 | 77.6                | 50.4                   | 17.3                  | 80.8                | 15.3                  | 28.8                 | 3.4                  | **40.7** | **43.6** |
> > >
> > > #### __References__
> > > [r1] Progressive graph learning for open-set domain adaptation.
> > > [r2] Weakly Supervised Open-set Domain Adaptation by Dual-domain Collaboration.
> > > [r3] Learning and the unknown: Surveying steps toward open world recognition.
> > > [r4] Towards Open Set Deep Networks.
> > > [r5] Open set domain adaptation by backpropagation.
> > > [r6] Open-Set Recognition: A Good Closed-Set Classifier is All You Need.
> > > [r7] Deep metric learning for open world semantic segmentation.

---

> ### Comment · Reviewer_r9Vn · 2023-01-31
> **Response to the authors**
>
> The reviewer thanks the response from the authors.
>
> Several concerns have been addressed, including ``effectiveness of unknown-class pseudo-label generation'', ``improvement of the proposed method'', and ``more recent  methods''.
>
> However, I still think the novelty is somewhat limited. In addition, the assumption for the unknown classes is still not convincing to me.
>
> Taking these two drawbacks, I would like to give a borderline to this paper and would like to see the opinions of the other reviewers.

---

> > ### Author Response · Authors · 2023-02-01
> > **Thanks for reviewing our paper**
> >
> > Dear Reviewer r9Vn,
> >
> > Many thanks for your valuable comments and reply. The quality of our paper has been improved because of your comments.
> >
> > Although we still have divergence in the novelty of this paper, we want to remind that the novelty or significance of methods is not the selection criteria of TMLR (https://jmlr.org/tmlr/acceptance-criteria.html).
> >
> > Now, our experiments are solid according to your reply (many thanks for your initial comments), and our research problem lies in the interests of the TMLR audience based on your initial comments.
> >
> > Best regards,
> >
> > Authors

---

### Review · Reviewer_2jSe · 2022-12-24

**Summary Of Contributions:**

This paper presents a method for cross-domain semantic segmentation that explicitly recognizes unknown classes in the target domain along with source-to-target transfer.  Two pixelwise classification heads are used, one of which classifies the known K source classes, while the other adds a K+1'th unit in addition that classifies the "unknown" category.  The predictions from the first are used to create online pseudolabel targets for the second, by targeting "unknown" for pixels with softmax prediction closest to the source class frequency prior, using EMA to find a value that thresholds a fixed budget fraction on average.  Interestingly, the pseudolabel bitmap is also used to disable the adversarial feature-discrimination loss used by the cross-domain segmentation systems, as these features must be kept different from the source in order to correctly classify as "unknown".  The system is integrated into three different segmentation systems, each evaluated using three synthetic driving datasets transferring to one real (cityscapes).  An additional application in a CT segmentation task for COVID-19 is also presented.


**Audience:**

Yes

**Claims And Evidence:**

No

**Requested Changes:**

Max softmax probability threshold for "unknown" class (MSP baseline)

Study on different values for fraction $\gamma$

Both of these are necessary additions in my view.


**Strengths And Weaknesses:**

This is a relatively simple system that results in decent but incremental gains, particularly in recognizing the unknown class, which the other segmentation systems are not capable of.  Even though it is straightforward, I still have questions on whether even simpler versions of this idea might be as effective, some of which should be included as additional baselines.  In particular, the post-hoc "unknown" classifier that simply thresholds on max softmax score should be included as a comparison.

There is also some related work that could be discussed more as well on open-set recognition in the context of classification (e.g. Vaze et al 2022 https://openreview.net/pdf?id=5hLP5JY9S2d, & related prior works) as well as anomaly detection (e.g. Roth et al 2021 https://arxiv.org/pdf/2106.08265.pdf), which produces low-resolution heatmaps for "unknown" class as well.  Note the "MSP" baseline used in Vaze et al is the post-hoc thresholding classifer I think should be a baseline.

In addition, I don't know that the source-to-target difference is necessary --- the same method could be applied to same-domain (source only), training on a split where there are held-out classes in the test set.  That would evaluate the effectiveness on unknown class shifts, not in the presence of target domain shift.  The same method could then be adapted also for cross-domain synthetic-to-real.  This could strengthen the paper by demonstrating integration of the method in multiple settings.



Additonal questions and comments:

* How is the p^prior calculated?  Is it the frequency of class pixels?  Or is it optimized in some other way?

    * If it is just a frequency count, the uniform distribution would be even simpler, and could be as effective --- is it?

    * A single vector for p^prior appears to assume an independent class distribution, but often class presence is correlated.  For example, residential areas have buildings, sidewalk, pedestrians, etc., while highways have more medians and overhead signs.  So the prior class distribution could also be modeled with a more sophisticated distribution, that has has nonzero covariance and/or is multimodal.  Is there any benefit to modeling this?

* Are the two classification heads C and C^* necessary?  An alternative would be to use just one head with K+1 units, but two softmaxes, one on all K+1 classes, and one on just the first K.  The second could be used in place of C^*, as it conditions on the pixel being "known".

* What is the effect of changing $\gamma$?  Particularly since the unknown class appears under-predicted, this would be interesting to vary.  Is "unknown" predicted more often on the target if this is increased?

* Sec 2 problem statement:  It would be good to also include the words "source" and "target in the definition here, when describing X^S ad X^T (e.g., "Let X^T be a r.v. following a distribution (called the "target" image data)" or along these lines, similarly for X^S and source).  Right now being presented with X^T at the start of the definition is easy to mistake the T for transpose.

* The citation format in text would be clearer including brackets around authors and year, not just year. Right now these can blend with the rest of the text

---

> ### Author Response · Authors · 2023-01-16
> **Response to Reviewer 2jSe (Part 1)**
>
> Thank you for your detailed review. we hope the following responses could address your concerns:
>
> #### __Q1: Add MSP as an additional baseline.__
> __A1:__  As seen in Table 4, we also use the other statistical metric (KL divergence instead of L2 norm). A common practice for identifying the unknown-class pseudo-labels is to compare the maximum softmax probability with a threshold (Luo et al., denoted as MSP). Therefore, it is a good suggestion to use "MSP" as a baseline. We conduct the ablation study and show the results in the following table:
>
> | Metrics                                 |      | road | side. | build.| fence | pole | sign | veg. | sky | person | rider | car | motor | bike | unk. | mIoU          | mIoU* |
> |-----------------------------------------|------|----------------------|-----------------------|------------------------|-----------------------|----------------------|----------------------|----------------------|---------------------|------------------------|-----------------------|---------------------|-----------------------|----------------------|----------------------|---------------|---------------|
> | MSP (Luo et al.) | B    | 81.9                 | 37.6                  | 77.8                   | 0.0                   | 19.4                 | 2.2                  | 77.8                 | 79.6                | 50.3                   | 18.2                  | 61.4                | 14.7                  | 34.3                 | 5.1                  | 40.0          | 42.7          |
> |                                         | L    | 82.2                 | 39.4                  | 76.2                   | 0.0                   | 18.1                 | 1.6                  | 75.7                 | 78.7                | 47.2                   | 16.9                  | 57.6                | 17.0                  | 28.6                 | 4.5                  | 38.8          | 41.5          |
> | L2 norm                  | B    | 82.4                 | 37.3                  | 76.4                   | 0.0                   | 22.1                 | 2.5                  | 76.6                 | 77.8                | 49.9                   | 18.5                  | 72.4                | 15.3                  | 28.9                 | **5.3**         | 40.4          | 43.1          |
> |                                         | L    | 80.2                 | 38.3                  | 76.9                   | 0.0                   | 20.3                 | 2.5                  | 76.8                 | 78.8                | 51.0                   | 17.7                  | 71.6                | 14.2                  | 27.9                 | **4.8**         | 40.1          | 42.8          |
> | KL divergence            | B    | 80.2                 | 37.4                  | 77.6                   | 0.1                   | 23.2                 | 2.8                  | 77.0                 | 78.5                | 49.4                   | 17.7                  | 80.4                | 15.0                  | 30.1                 | 3.9                  | **40.9** | 	**43.8** |
> |                                         | L    | 78.8                 | 38.6                  | 77.4                   | 0.0                   | 22.9                 | 2.5                  | 76.6                 | 77.6                | 50.4                   | 17.3                  | 80.8                | 15.3                  | 28.8                 | 3.4                  | **40.7** | **43.6** |
>
> We can see that the L2 norm outperforms better than MSP in terms of both segmenting unknown class (unk.) and known classes (mIoU*), thus a higher overall segmentation ability (mIoU). Although KL divergence obtains relatively lower unknown-class IoU, it has superior performance in segmenting known classes (mIoU*) and the best segmentation overall performance. Both L2 norm and KL divergence leverage the property of the unknown class to compare the output probability distribution of a pixel in the test image with the known-class prior probability distribution. However, MSP does not utilize the property of the unknown class derived from the source data, which may be the main reason why MSP achieves an inferior performance.
>
> #### __Q2: Evaluate the effectiveness on same-domain setting (i.e., only unknown class shifts without target domain shifts).__
> __A2:__ Thanks for your inspiration for extending our method in more settings. We have conducted experiments to prove that the method could be applied to same-domain (source only), i.e., in the presence of unknown class shifts without target domain shifts:

---

> > ### Author Response · Authors · 2023-01-16
> > **Response to Reviewer 2jSe (Part 2)**
> >
> > #### __Table: Results of CLAN + KRADA applied to the same domain (SYNTHIA → SYNTHIA with the unknown class).__
> > | Metrics   |      | road | side. | build.| fence | pole | sign | veg. | sky | person | rider | car | motor | bike | unk. | mIoU          | mIoU* |
> > |-----------------------------------------|------|----------------------|-----------------------|------------------------|-----------------------|----------------------|----------------------|----------------------|---------------------|------------------------|-----------------------|---------------------|-----------------------|----------------------|----------------------|---------------|---------------|
> > | Same domain | B | 75.9                 | 44.5                  | 80.0                   | 26.6                  | 39.72                | 0.2                  | 60.3                 | 81.3                | 54.3                   | 41.6                  | 60.6                | 23.7                  | 28.6                 | 8.6                  | 44.7 | 47.5          |
> > |                            | L | 69.7                 | 44.4                  | 85.0                   | 27.0                  | 39.5                 | 0.1                  | 63.0                 | 81.8                | 56.1                   | 36.7                  | 57.5                | 17.6                  | 24.8                 | 7.7                  | 43.6 | 46.4          |
> > | OSDAS       | B | 82.4                 | 37.3                  | 76.4                   | 0.0                   | 22.1                 | 2.5                  | 76.6                 | 77.8                | 49.9                   | 18.5                  | 72.4                | 15.3                  | 28.9                 | 5.3                  | 40.4 | 43.1          |
> > |                            | L | 80.2                 | 38.3                  | 76.9                   | 0.0                   | 20.3                 | 2.5                  | 76.8                 | 78.8                | 51.0                   | 17.7                  | 71.6                | 14.2                  | 27.9                 | 4.8                  | 40.1 | 42.8          |
> >
> > We can see that the proposed method not only can be applied to OSS but also is applicable to the same-domain setting (i.e., only unknown class shifts without target domain shifts).
> >
> > #### __Q3: About the $p^{prior}$.__
> > __A3:__  The $p^{prior}$ is a vector that is calculated by counting the frequency of class pixels derived from the source data and then normalized. We also used the uniform distribution to depict the distribution characteristics of the unknown class, and the results are shown in the bellow table:
> >
> > __Table: Results of CLAN + KRADA with different $p^{prior}$ for the unknown-class pseudo-label generation on SYNTHIA → Cityscapes.__
> > | Metrics   |      | road | side. | build.| fence | pole | sign | veg. | sky | person | rider | car | motor | bike | unk. | mIoU          | mIoU* |
> > |-----------------------------------------|------|----------------------|-----------------------|------------------------|-----------------------|----------------------|----------------------|----------------------|---------------------|------------------------|-----------------------|---------------------|-----------------------|----------------------|----------------------|---------------|---------------|
> > | Uniform distribution           | B | 80.6                 | 36.8                  | 77.8                   | 0.0                   | 24.6                 | 2.4                  | 76.7                 | 77.1                | 49.3                   | 18.1                  | 81.7                | 14.3                  | 30.7                 | 2.1                  | __40.9__ | __43.9__ |
> > |                                               | L | 77.6                 | 37.4                  | 77.0                   | 0.0                   | 22.4                 | 2.0                  | 76.2                 | 75.2                | 48.5                   | 17.3                  | 80.1                | 13.8                  | 29.8                 | 1.6                  | 39.9          | 42.9          |
> > | Known-class prior distribution | B | 80.2                 | 37.4                  | 77.6                   | 0.1                   | 23.2                 | 2.8                  | 77.0                 | 78.5                | 49.4                   | 17.7                  | 80.4                | 15.0                  | 30.1                 | __3.9__         | __40.9__ | 43.8          |
> > |                                               | L | 78.8                 | 38.6                  | 77.4                   | 0.0                   | 22.9                 | 2.5                  | 76.6                 | 77.6                | 50.4                   | 17.3                  | 80.8                | 15.3                  | 28.8                 | __3.4__        | __40.7__| __43.6__ |

---

> > > ### Author Response · Authors · 2023-01-16
> > > **Response to Reviewer 2jSe (Part 3)**
> > >
> > > Although the uniform distribution achieves the similar mIoU and mIoU* as the known-class prior distribution at the best epoch, its unknown-class IoU values are significantly lower at the both last and best epochs. This demonstrates that the known-class prior distribution is more suitable to portray the distribution characteristics of the unknown class while taking full advantage of the data attribute of source data.
> > >
> > > Let us explain from another viewpoint. As we all know, it is hard to accurately depict the unknown classes, due to the complexity and diversity of “unknown” classes in the real world. A plain and ideal condition that an unknown class is independent of known classes, and its prediction probabilities could not bias any known classes ideally, i.e., its distribution conforms to a uniform distribution. But this assumption is very strong and ideal and means that we have no any prior knowledge about the target domain. Benefiting from the prior knowledge of source data, we can relax this assumption by utilizing the class prior probability of source classes. Although this distribution property is not very perfect to depict the unknown classes in all cases world (as the examples the reviewer mentioned), experimental results verify that this assumption is rational and feasible. Besides, we also noticed that there is less research to explore what the unknown class and its essential characteristics are. We hope our work can give inspiration to model the unknown class and a more sophisticated distribution deserves further in-depth exploration.
> > >
> > > #### __Q4: About the two classification heads C and C*.__
> > > __A4:__ Two classification heads C and C* are necessary, since these two classification heads have different functions and don’t mix with each other. Specifically, C* aims to generate unknown-class pseudo-labels by outputting prediction probability over known classes (K classes) and this pseudo-label generation process should get rid of the effects of the unknown class. C is open-set pixel-level classifier to output the softmax probability over K+1 classes. Experimental results shown below also demonstrate that two classification heads are necessary.
> > > | Metrics                                 |      | road | side. | build.| fence | pole | sign | veg. | sky | person | rider | car | motor | bike | unk. | mIoU          | mIoU* |
> > > |-----------------------------------------|------|----------------------|-----------------------|------------------------|-----------------------|----------------------|----------------------|----------------------|---------------------|------------------------|-----------------------|---------------------|-----------------------|----------------------|----------------------|---------------|---------------|
> > > | C                | B | 67.3                 | 36.0                  | 76.1                   | 0.0                   | 20.6                 | 2.1                  | 72.1                 | 73.5                | 48.5                   | 16.3                  | 64.9                | 11.9                  | 28.3                 | 3.5                  | 37.2          | 39.8          |
> > > |                                 | L | 55.7                 | 31.3                  | 75.5                   | 0.0                   | 18.3                 | 1.8                  | 75.1                 | 76.8                | 48.7                   | 17.0                  | 62.7                | 13.7                  | 30.0                 | 2.9                  | 36.4          | 39.0          |
> > > | C and C* | B | 82.4                 | 37.3                  | 76.4                   | 0.0                   | 22.1                 | 2.5                  | 76.6                 | 77.8                | 49.9                   | 18.5                  | 72.4                | 15.3                  | 28.9                 | __5.3__        | __40.4__| __43.1__ |
> > > |                                 | L | 80.2                 | 38.3                  | 76.9                   | 0.0                   | 20.3                 | 2.5                  | 76.8                 | 78.8                | 51.0                   | 17.7                  | 71.6                | 14.2                  | 27.9                 | __4.8__         | __40.1__ | __42.8__|
> > >
> > >
> > > #### __Q5: About the effect of changing γ.__
> > > __A5:__ We take full advantage of source data to achieve an adaptive threshold δ. Specifically, we also calculate the L2 norm distance between pprior and the softmax probabilities of source images outputted by C*. Since source images do not include any unknown classes, the threshold δ should ensure that the number of pixels in the source images which are wrongly recognized as unknown is not too much. Therefore, γ is such a proportion to represent the tolerable ratio of pixels in source images wrongly recognized into the unknown class. γ should be a small value. We change the value of γ and show the results in the bellow table:

---

> > > > ### Author Response · Authors · 2023-01-16
> > > > **Response to Reviewer 2jSe (Part 4)**
> > > >
> > > > #### __Table: Results of CLAN + KRADA with varying γ on SYNTHIA → Cityscapes.__
> > > > | Metrics                                 |      | road | side. | build.| fence | pole | sign | veg. | sky | person | rider | car | motor | bike | unk. | mIoU          | mIoU* |
> > > > |--------------------------------|---|------|------|------|-----|------|-----|------|------|------|------|------|------|------|--------------|---------------|---------------|
> > > > | 0.005%         | B | 79.8 | 38.2 | 77.3 | 0.0 | 22.1 | 2.3 | 76.9 | 81.0 | 51.9 | 19.3 | 78.8 | 20.7 | 28.1 | 0.1          | __41.2__ | __44.4__ |
> > > > |                                | L | 82.1 | 37.5 | 77.8 | 0.0 | 24.4 | 2.5 | 78.0 | 78.4 | 46.5 | 18.8 | 80.9 | 13.1 | 28.2 | 0.1          | __40.6__| __43.7__ |
> > > > | 0.01%          | B | 80.3 | 39.4 | 78.2 | 0.1 | 24.8 | 2.8 | 78.0 | 79.3 | 49.4 | 18.0 | 80.8 | 12.9 | 31.0 | 1.6          | __41.2__ | 44.2          |
> > > > |                                | L | 81.0 | 38.6 | 78.0 | 0.0 | 24.0 | 2.5 | 78.0 | 76.8 | 47.7 | 17.5 | 79.6 | 11.3 | 29.8 | 1.0          | 40.4          | 43.4          |
> > > > | 0.05% | B | 82.4 | 37.3 | 76.4 | 0.0 | 22.1 | 2.5 | 76.6 | 77.8 | 49.9 | 18.5 | 72.4 | 15.3 | 28.9 | __5.3__ | 40.4          | 43.1          |
> > > > |                                | L | 80.2 | 38.3 | 76.9 | 0.0 | 20.3 | 2.5 | 76.8 | 78.8 | 51.0 | 17.7 | 71.6 | 14.2 | 27.9 | __4.8__| 40.1          | 42.8          |
> > > > | 0.1%           | B | 79.0 | 35.8 | 76.6 | 0.0 | 21.4 | 1.8 | 76.5 | 76.5 | 47.4 | 17.7 | 72.9 | 14.2 | 27.2 | 4.6          | 39.4          | 42.1          |
> > > > |                                | L | 73.9 | 32.0 | 73.9 | 0.0 | 19.3 | 1.9 | 74.4 | 77.0 | 47.5 | 16.2 | 61.2 | 11.8 | 24.9 | 4.0          | 37.0          | 39.5          |
> > > > | 0.5%           | B | 70.3 | 31.9 | 73.7 | 0.1 | 19.7 | 0.9 | 69.8 | 71.9 | 48.9 | 18.3 | 64.4 | 16.5 | 33.6 | 3.9          | 37.4          | 40.0          |
> > > > |                                | L | 49.9 | 11.3 | 48.0 | 0.0 | 10.2 | 0.1 | 34.6 | 45.4 | 27.1 | 7.5  | 28.7 | 6.0  | 8.5  | 2.1          | 20.0          | 21.3          |
> > > >
> > > > Compared with 0.05% (the value reported in the paper), reducing γ (0.005% and 0.01%) leads to the decrease of identifying the unknown-class pixels (lower unknown-class IoU) and the increase of segmenting known classes and overall performance (mIoU* and mIoU). On the contrary, increasing γ(0.1% and 0.5%) does not mean a higher unknown-class IoU and instead, it diminishes the known-class segmentation performance and thus an inferior overall performance (lower mIoU* and mIoU). More specifically, we achieve unsatisfactory and unstable results when γ is set as 0.5%.  We choose γ between 0.01% ~ 0.1% to obtain satisfactory and competitive segmentation performance.
> > > >
> > > > #### __Q6: About the Sec 2 problem statement and citation format.__
> > > > __A6:__ Thanks for your advice about the Sec 2 problem statement and citation format in the text. We have revised them to clear up the confusion.
> > > >
> > > > #### __References__
> > > > [r1] Open-Set Recognition: A Good Closed-Set Classifier is All You Need, Vaze et al., ICLR 2021.
> > > > [r2] Progressive graph learning for open-set domain adaptation, Luo et al., ICML 2020.

---

### Review · Reviewer_biFQ · 2023-01-03

**Summary Of Contributions:**

This paper present a method for open world semantic segmentation in which the method is not only able to distinguish pixels coming from new categories, but also align the classifier to the deployment data that might have a different distribution that the data used for training.
The contributions of the paper are the following:
- this method solves the problem of open-world semantic segmentation, in which there can be new categories not seen during training and the alignment problem, in which even the categories seen during training they might be affected by a domain shift.
- The method is evaluated on two synthetic to real segmentation datasets for autonomous driving and one for detecting COVID19 from CT scans.


**Audience:**

Yes

**Broader Impact Concerns:**

No broader impact concerns found.

**Claims And Evidence:**

No

**Requested Changes:**

- (Critical) The presentation of the proposed work is misleading in my opinion. the term open world semantic segmentation (OSS) as the task they aim to solve. However, the term OSS normally does not consider the domain alignment. Thus, I think that the authors should propose another name for their task in which is clear that they do unknown class detection and domain alignment jointly.

- (Critical) In the introduction authors present only some papers for closed-set domain adaptation for semantic segmentation. To have a better idea of what has been already proposed, the authors should provide a section of related work in which they distinguish their settings from previous work in related topics, for instance open world approaches and domain adaptation methods for image classification and segmentation. This would provide also better and more interesting baselines to compare with.

- (Critical) The definition in problem 1 is difficult to read and understand and it should be improved. Authors first talk about target domain $X^T$, then they go back to source domain $X^T$ and then back again to target, while presenting the definition of some variables in the wrong location. Also, in most of the cases authors use "\mathcal" for sets, but $\mathcal M$ is used for the model.

- (Critical) In several parts of the paper (e.g. end of 1st paragraph section 4.1, remark 2) authors mention that they highly fitted pixels are classified as unknown pixels. In my understanding it should be the other way around. Only low-fitting pixels should be classified as unknown. Authors should clarify this point and if needed, correct it.

- (Critical) The evaluation is performed by comparing the proposed approach that learn to select which pixels come from unknown distributions to methods for closed-set domain adaptation (excluding Saito et al. 2018b). It is evident that the proposed method works better than simple domain adaptation approaches when the target domain has new classes, as that is the aim of the proposed method. What would happen when there are no new classes?

- (Critical) The evaluation of the method is done by explicitly removing images containing certain classes form a given training dataset. How are those classes chosen? To have a better evaluation of the proposed approach authors should evaluate the average performance  when different classes are removed (Not only one additional case in the ablation study).

- (Minor) At the end of page 1, authors talk about variations in light for domain shift in CT scans. In that case, in my understanding there is no light, as we are not dealing with natural images.

- (Minor) Not sure what is the advantage of presenting results for the best models (B) and the last iteration model (L). I guess, this evaluation come from previous work, but authors should explain the reason or present only the best model which makes more sense to me.



**Strengths And Weaknesses:**

\+ Soling the two task jointly makes sense as It is very likely that the two problems occur when dealing with open world scenario.

\- However, this task was already proposed by (Saito et al. ECCV18) for classification. Extending it for semantic segmentation does not seem an important contribution in my opinion.

\- In general the presentation of the paper is ok, but there are some parts that are not clear or misleading and should be corrected. (see next section)

\- In the beginning of the second page, the authors present the term open world semantic segmentation (OSS) as it is a term that they invented. Instead there are already multiple papers talking about OSS that should be cited. Even more, in this paper, the new classes are just detected as outliers, while in other works (e.g. Cen et al. ICCV21) the method not only detects pixels of new classes, but it is also able to cluster them in meaningful categories. In general there is some confusion with the term open world. For instance, in my understanding when talking about open world I would assume that there is no knowledge at all of the deployment domain, as in domain generalization.

---

> ### Author Response · Authors · 2023-01-16
> **Response to Reviewer biFQ (Part 1)**
>
> Thank you for your review. we hope the following responses could address your concerns:
>
> #### __Q1: The term OSS is misleading.__
> __A1:__  Thanks for this suggestion. We have revised the name of the setting (OSS) into open-set domain adaptation segmentation (OSDAS) to highlight both the unknown class detection and domain alignment.
>
> #### __Q2: About the related works and the comparison with other works.__
> __A2:__  In fact, there is no relevant work to solve the scenario that is completely consistent with ours: open set, domain adaptation, and semantic segmentation. Therefore, we have reviewed two kinds of much more relevant related works: 1) unsupervised open set domain adaptation (UOSDA) classification and 2) closed set domain adaptation for semantic segmentation (CSDAS) in the supplementary material. We reviewed a group of works about UOSDA and explicitly pointed out the disadvantages of these existing UOSDA methods which only focus on classification tasks and most of them are not easily modified for semantic segmentation. Therefore, exploring the open set domain adaptation in semantic segmentation is challenging and valuable.
>
> Although the open set domain adaptation setting is proposed by Saito et al., experimental results demonstrate that simply extending OSBP (a representative UOSDA method proposed by Saito et al.,) to semantic segmentation achieves poor performance (seen in Table 1, 2, and 5).
>
> In addition, we have also distinguished the proposed setting from the existing settings in previous works in Section 2 (definition of the proposed setting).
>
> We recently found that Cen et al. solves a similar open-set semantic segmentation setting in [r7], but they did not care about the distribution shift between source data and target data. In addition, Cen et al. adopted a few-shot learning method that needs labels to give additional annotations. Their work is very different from ours and we have to try to find the comparable part which is the unknown-class pseudo-label generation method.
> Therefore, we compared the performance of different unknown-class pseudo-label generation criteria: including the MSP in Luo et al. [r1], DML in Cen et al. [r7], and show the results in the bellow table:

---

> > ### Author Response · Authors · 2023-01-16
> > **Response to Reviewer biFQ (Part 2)**
> >
> > #### __Table 10: Comparison with other related works.__
> >
> > | Metrics                                 |      | road | side. | build.| fence | pole | sign | veg. | sky | person | rider | car | motor | bike | unk. | mIoU          | mIoU* |
> > |-----------------------------------------|------|----------------------|-----------------------|------------------------|-----------------------|----------------------|----------------------|----------------------|---------------------|------------------------|-----------------------|---------------------|-----------------------|----------------------|----------------------|---------------|---------------|
> > | MSP (Luo et al.) | B    | 81.9                 | 37.6                  | 77.8                   | 0.0                   | 19.4                 | 2.2                  | 77.8                 | 79.6                | 50.3                   | 18.2                  | 61.4                | 14.7                  | 34.3                 | 5.1                  | 40.0          | 42.7          |
> > |                                         | L    | 82.2                 | 39.4                  | 76.2                   | 0.0                   | 18.1                 | 1.6                  | 75.7                 | 78.7                | 47.2                   | 16.9                  | 57.6                | 17.0                  | 28.6                 | 4.5                  | 38.8          | 41.5          |
> > | DML (Cen et al.) | B    | 77.3                 | 37.4                  | 77.3                   | 0.0                   | 23.9                 | 1.6                  | 75.9                 | 68.6                | 48.4  | 16.5                 | 76.7                  | 12.8                   | 30.7                  | 0.9                  | 39.1                 | 42.1                 |
> > |                                         | L    | 72.6                 | 35.9                  | 77.3                   | 0.0                   | 22.0                 | 1.7                  | 71.4                 | 62.7                | 47.3                   | 18.5                  | 79.8                | 9.7                   | 37.3                 | 1.4                  | 38.4          | 41.3          |
> > | MSP* (Vaze et al.) | B    | 82.5                 | 35.1                    | 75.0                      | 0.0                   | 19.7                 | 2.4                  | 74.7               | 77.2                | 47.6                  | 19.3                  | 78.9                | 7.0                  | 29.2                 | 2.5                  | 39.4          | 42.2          |
> > |                                         | L    | 79.5                 | 35.6                     | 73.6                     | 0.0                   | 17.4                 | 2.1                  | 74.1                 | 76.3                | 48.4                   | 19.0                  | 73.8                | 7.5                  | 29.7                 |2.2                  | 38.5          | 41.3         |
> > | L2 norm                  | B    | 82.4                 | 37.3                  | 76.4                   | 0.0                   | 22.1                 | 2.5                  | 76.6                 | 77.8                | 49.9                   | 18.5                  | 72.4                | 15.3                  | 28.9                 | **5.3**         | 40.4          | 43.1          |
> > |                                         | L    | 80.2                 | 38.3                  | 76.9                   | 0.0                   | 20.3                 | 2.5                  | 76.8                 | 78.8                | 51.0                   | 17.7                  | 71.6                | 14.2                  | 27.9                 | **4.8**         | 40.1          | 42.8          |
> > | KL divergence            | B    | 80.2                 | 37.4                  | 77.6                   | 0.1                   | 23.2                 | 2.8                  | 77.0                 | 78.5                | 49.4                   | 17.7                  | 80.4                | 15.0                  | 30.1                 | 3.9                  | **40.9** | 	**43.8** |
> > |                                         | L    | 78.8                 | 38.6                  | 77.4                   | 0.0                   | 22.9                 | 2.5                  | 76.6                 | 77.6                | 50.4                   | 17.3                  | 80.8                | 15.3                  | 28.8                 | 3.4                  | **40.7** | **43.6** |
> >
> > Our proposed methods (L2 norm and KL divergence) significantly outperform MSP [r1], DML [r7], and MSP* [r6]. Detailed descriptions of this table can be seen in the newly submitted manuscript.
> >
> > #### __Q3: About the misleading definition in problem 1.__
> > __A3:__ We have revised the definition in problem 1 to clean up the confusion.

---

> > > ### Author Response · Authors · 2023-01-16
> > > **Response to Reviewer biFQ (Part 3)**
> > >
> > > #### __Q4: About the description of “highly fitted pixels”.__
> > > __A4:__ A common practice to generate unknown-class pseudo-labels is to compare the maximum softmax probability or entropy with a threshold and mark the data with high uncertainty as an unknown class [r1, r2].  In principle, this practice selects pixels with high prediction uncertainty or low confidence as unknown classes, which may be the reason why the reviewer thinks about the low-fitting pixels. Nevertheless, it should be noted that uncertainty is not unknown [r3], and thresholding on uncertainty is not sufﬁcient to determine what is unknown [r4].
> > >
> > > In this work, we aim to explore what is unknown and explicitly point out the characteristic of the unknown class which is less discussed and noticed in the current works. Benefiting from the prior knowledge of source data, we propose to use the class prior probability of source classes to model the unknown class and assume an unknown pixel would conform to a known-class prior probability distribution. Therefore, we need to select the pixels whose softmax distributions are close to the known-class prior probability distribution, i.e., highly-fitting pixels, as the unknown-class pseudo-labels.
> > >
> > > #### __Q5: About the case when there are no new classes.__
> > > __A5:__ In this work, we aim to propose a quite simple and general approach based on the well-studied closed set domain adaptation for semantic segmentation (CSDAS) methods, so that we can make a few modifications to adapt CSDAS methods for the open world setting. We have realized KRADA on three different CSDAS methods and experimental results show the effectiveness and generalization ability of the proposed KRADA in identifying the unknown class and known-class domain alignment. If there is no new class in target data, it becomes a closed set domain adaptation segmentation task which can be well solved by current CSDAS methods.
> > >
> > > #### __Q6: About the simulation of the unknown class.__
> > > __A6:__ There is no standard dataset for such a new setting, so we have to simulate the open-set scenario based on common datasets in the closed-set setting (CSDAS): SYNTHIA → Cityscapes and GTA5 → Cityscapes. Specifically, we choose several classes as the unknown class and discard those source images containing the pixels belonging to the unknown class. To have enough eligible source images left, we tend to select the minority classes to form the unknown class. But we also observed that this may be the main reason why the obtained unknown-class IoU is so small, especially in Table 2. Then we changed the composition of the unknown class and conducted an extra ablation study in Table 3. These two cases are sufficient and representative to evaluate the performance of the proposed KRADA. It is difficult to consider all different categories as unknown, since there are no or few source images left for some cases. Besides, the number of all the cases is a large combination number ($C_{cls}^1$  +  $C_{cls}^3$ +  $C_{cls}^3$ +…, where $cls$ denotes the number of known classes).
> > >
> > > #### __Q7: About the description of “variations in light for domain shift in CT scans”.__
> > > __A7:__ Here we talk about domain shifts in a general condition, not only the variations in CT scans. Just take CT scans as an example. By the way, the CT scan acquisition process needs X-rays which belong to a kind of high-frequency invisible light.
> > >
> > > #### __Q8: About reporting results at both the best (B) and the last (L) models.__
> > > __A8:__ It is comprehensive and objective to evaluate the performance and stability of the models by reporting both the best and last scores, not only showing the best performance, like [r8].
> > >
> > > #### __References__
> > > [r1] Progressive graph learning for open-set domain adaptation, Luo et al., ICML 2020.
> > > [r2] Weakly Supervised Open-set Domain Adaptation by Dual-domain Collaboration, Tan et al., CVPR 2019.
> > > [r3] Learning and the unknown: Surveying steps toward open world recognition, Boult et al., AAAI 2019.
> > > [r4] Towards Open Set Deep Networks, Bendale et al., CVPR 2016.
> > > [r5] Open set domain adaptation by backpropagation, Saito et al., ECCV 2018.
> > > [r6] Open-Set Recognition: A Good Closed-Set Classifier is All You Need, Vaze et al., ICLR 2021.
> > > [r7] Deep metric learning for open world semantic segmentation. Cen et al., ICCV 2021.
> > > [r8] Unsupervised Label Noise Modeling and Loss Correction. Arazo et al., International conference on machine learning. PMLR, 2019.

---

### Decision · Action_Editors · 2023-02-13

**Recommendation:** Accept as is

**Comment:**

Thanks for your submission to TMLR and for the edits made to the paper based on the reviewer comments.  Overall, two of the three reviewers were quite positive about the paper, particularly after the discussion phase.  One reviewer had some concerns about novelty but, as pointed out by the authors, these concerns do not weigh heavily on the final decision.  Overall it does seem that the technical concerns have been addressed, and the problem studied would be of interest to several in the community.  I'm happy to recommend accepting the paper.

**Audience:**

Yes, this paper definitely would be of interest to several people in the TMLR community (myself included).

**Claims And Evidence:**

Yes, the claims are supported by clear evidence, particularly after the discussion with the reviewers, where additional empirical results were added and the method was further clarified.

---

> ### Author Response · Authors · 2023-02-14
> **Many thanks for your acknowledgement**
>
> Dear EIC, handling AE and Reviewers,
>
> Many thanks for your acknowledgement, support and constructive discussions!! Moreover, we want to thank AE and three reviewers again for their active engagement: the constructive discussions have definitely helped improve our draft a lot. We will provide camera ready revision and opensource our research code soon.
>
>
> Best regards,
>
> Authors of Paper 654